# Accelerated Multiple Wasserstein Gradient Flows for Multi-objective Distributional Optimization

**Dai Hai Nguyen** [1]  **Dung Duc Nguyen** [2]  **Atsuyoshi Nakamura** [1]  **Hiroshi Mamitsuka** [3]

## Abstract

We study multi-objective optimization over probability distributions in Wasserstein space. Recently, Nguyen et al. (2025) introduced Multiple Wasserstein Gradient Descent (MWGraD) algorithm, which exploits the geometric structure of Wasserstein space to jointly optimize multiple objectives. Building on this approach, we propose an accelerated variant, A-MWGraD, inspired by Nesterov's acceleration. We analyze the continuous-time dynamics and establish convergence to weakly Pareto optimal points in probability space. Our theoretical results show that A-MWGraD achieves a convergence rate of $\mathcal{O}(1/t^2)$ for geodesically convex objectives and $\mathcal{O}(e^{-\sqrt{\beta}t})$ for $\beta$-strongly geodesically convex objectives, improving upon the $\mathcal{O}(1/t)$ rate of MWGraD in the geodesically convex setting. We further introduce a practical kernel-based discretization for A-MWGraD and demonstrate through numerical experiments that it consistently outperforms MWGraD in convergence speed and sampling efficiency on multi-target sampling tasks.

## 1. Introduction

We study multi-objective optimization over the space of probability distributions. Let $F_1, F_2, \ldots, F_K : \mathcal{P}_2(\mathcal{X}) \rightarrow \mathbb{R}$ be a set of $K$ objective functionals ($K \geq 2$) defined on the space $\mathcal{P}_2(\mathcal{X})$ of probability distributions on $\mathcal{X} \subseteq \mathbb{R}^d$ with finite second-order moment. Our goal is to find an optimal distribution $\rho \in \mathcal{P}_2(\mathcal{X})$ that minimizes the vector-valued objective

$$\min_{\rho \in \mathcal{P}_2(\mathcal{X})} \mathbf{F}(\rho) = \min_{\rho \in \mathcal{P}_2(\mathcal{X})} (F_1(\rho), F_2(\rho), ..., F_K(\rho)) . \quad (1)$$

This problem, known as *Multi-Objective Distributional Optimization* (MODO, Nguyen et al. (2025a)), can be regarded as a natural extension of classical multi-objective optimization (MOO, Sawaragi et al. (1985)) from the Euclidean spaces to the probability spaces (or spaces of probability distributions). A motivating example is multi-target sampling (Liu et al., 2021; Phan et al., 2022), where the goal is to generate samples or particles that simultaneously approximate multiple unnormalized target distributions. In this setting, each objective corresponds to the Kullback–Leibler (KL) divergence between the current distribution and one of the targets. To solve MODO, Nguyen et al. (2025a) recently proposed the *Multiple Wasserstein gradient descent* (MWGraD) algorithm, which constructs a flow of distributions that gradually decreases all the objectives. At each iteration, it (i) estimates the Wasserstein gradient of each objective $F_k$, for $k \in \{1, 2, \ldots, K\}$, and (ii) aggregates them into a single Wassertein descent direction to update the particles.

In Euclidean optimization, standard gradient descent (GD) is often suboptimal, whereas momentum-based methods such as Nesterov's accelerated gradient (NAG, Nesterov (2013)) achieve improved convergence rates. Its continuous-time counterpart, known as the *accelerated gradient flow* (Su et al., 2016), has been extensively studied, including in multi-objective settings (Attouch & Garrigos, 2015; Sonntag & Peitz, 2024). In the context of optimization over probability spaces, Taghvaei & Mehta (2019) introduced accelerated flows from an optimal control perspective, while Wang & Li (2022) developed a unified framework of accelerated gradient flows in probability spaces under different information metrics, including Fisher-Rao, Wasserstein, Kalman-Wasserstein, and Stein metrics. Both lines of work interpret acceleration through the lens of damped Hamiltonian dynamics.

Given the success of accelerated methods, a natural question arises: *Can MWGraD be similarly accelerated in the probability-space settings?* In this work, we answer this question affirmatively. Our contributions are summarized as follows.

[1]Graduate School of Information Science and Technology, Hokkaido University, Japan [2]Institute of Information Technology, Vietnam Academy of Science and Technology, Vietnam [3]Bioinformatics Center, Kyoto University, Japan. Correspondence to: Dai Hai Nguyen <hai@ist.hokudai.ac.jp or haidnguyen0909@gmail.com>.

*Proceedings of the 43rd International Conference on Machine Learning*, Seoul, South Korea. PMLR 306, 2026. Copyright 2026 by the author(s).

1. We construct a continuous-time flow, termed the *MWGraD flow*, to analyze the convergence of MWGraD (Nguyen et al., 2025a) in the continuous-time limit. We define the merit function $\mathcal{M}(\rho) = \sup_{q \in \mathcal{P}_2(\mathcal{X})} \min_{k \in [K]} \{F_k(\rho) - F_k(q)\}$, where $[K] := \{1, 2, \ldots, K\}$, which has been used in prior analyses of multi-objective gradient flows in Euclidean spaces (Attouch & Garrigos, 2015; Sonntag & Peitz, 2024). In Theorem 3.5, we show that **MWGraD flow achieves a convergence rate of** $\mathcal{M}(\rho_t) = \mathcal{O}(1/t)$.

2. We design an accelerated variant, *A-MWGraD*, motivated by the perspective of damped Hamiltonian dynamics for Nesterov's acceleration, and prove in Theorem 3.7 faster convergence rates: $\mathcal{M}(\rho_t) = \mathcal{O}(1/t^2)$ for geodesically convex objectives and $\mathcal{M}(\rho_t) = \mathcal{O}(e^{-\sqrt{\beta}t})$ for $\beta$-strongly geodesically convex objectives. To the best of our knowledge, this is the **first acceleration result for multi-objective optimization in probability spaces**.

3. We develop a **sampling-efficient discrete-time implementation of the accelerated flow**, incorporating practical strategies for efficiently estimating Wasserstein gradients of objectives in multi-target sampling tasks. Experiments on synthetic and real-world datasets demonstrate that A-MWGraD consistently converges faster than MWGraD, validating our theoretical findings in practice.

## 2. Preliminaries

We begin by reviewing standard gradient flows in Euclidean and probability spaces, and then briefly introduce MODO and relevant background.

### 2.1. Gradient Flows on Euclidean space $\mathcal{X}$ and Their Acceleration

Given a differentiable function $f : \mathcal{X} \to \mathbb{R}$, the Euclidean gradient flow of $f$ is

$$\dot{\mathbf{x}}_t = -\nabla f(\mathbf{x}_t), \qquad (2)$$

where $\nabla f$ denotes the Euclidean gradient. Discretizing (2) using a standard Euler integrator yields GD algorithm:

$$\mathbf{x}_{n+1} = \mathbf{x}_n - \eta_n \nabla f(\mathbf{x}_n),$$

with step size $\eta_n > 0$. GD is known to be suboptimal; other first-order methods achieve better convergence guarantees and practical performance (Nemirovskij & Yudin, 1983; Nesterov, 2013). A key feature of many accelerated methods is the incorporation of *momentum*. For example,

Nesterov's Accelerated Gradient (NAG) method (Nesterov, 2013) updates according to

$$\mathbf{x}_n = \mathbf{m}_{n-1} - \eta_n \nabla f(\mathbf{m}_{n-1}),$$
$$\mathbf{m}_n = \mathbf{x}_n + \alpha_n(\mathbf{x}_n - \mathbf{x}_{n-1}),$$

where $\alpha_n > 0$ depends on the convexity of $f$. For $L$-smooth and $\beta$-strongly convex $f$, $\alpha_n = (\sqrt{L} - \sqrt{\beta})/(\sqrt{L} + \sqrt{\beta})$; otherwise, $\alpha_n = (n-1)/(n+2)$. The continuous-time limit of NAG satisfies an ordinary differential equation (ODE), known as the *accelerated gradient flow* (AGF) (Su et al., 2016; Shi et al., 2022)

$$\ddot{\mathbf{x}}_t + \alpha_t \dot{\mathbf{x}}_t + \nabla f(\mathbf{x}_t) = 0, \qquad (3)$$

where $\alpha_t = 2\sqrt{\beta}$ for $\beta$-strongly convex $f$ and $\alpha_t = 3/t$ for general convex $f$. Importantly, AGF (3) can be formulated as a damped Hamiltonian flow (Maddison et al., 2018)

$$\dot{\mathbf{x}}_t = \nabla_m H(\mathbf{x}_t, \mathbf{m}_t),$$
$$\dot{\mathbf{m}}_t = -\alpha_t \mathbf{m}_t - \nabla_{\mathbf{x}} H(\mathbf{x}_t, \mathbf{m}_t),$$

where $\mathbf{x}$ and $\mathbf{m}$ are the state and momentum variables, respectively, and $H(\mathbf{x}, \mathbf{m}) = f(\mathbf{x}) + \frac{1}{2}\|\mathbf{m}\|_2^2$ is the Hamiltonian. Thus, AGF can be interpreted as introducing a linear momentum term into the Hamiltonian dynamics.

### 2.2. Gradient Flows on Probability Spaces $\mathcal{P}_2(\mathcal{X})$ and Their Acceleration

Optimization over the probability space $\mathcal{P}_2(\mathcal{X})$ follows a similar principle. Given a functional $F : \mathcal{P}_2(\mathcal{X}) \to \mathbb{R}$, we require an analogue of the Euclidean gradient. The resulting gradient flow depends on the choice of metric on $\mathcal{P}_2(\mathcal{X})$.

**Definition 2.1** (Metric Tensor). Let $\rho \in \mathcal{P}_2(\mathcal{X})$. The tangent space at $\rho$ is $\mathcal{T}_\rho \mathcal{P}_2(\mathcal{X}) = \{\sigma \in \mathcal{F}(\mathcal{X}) : \int \sigma d\mathbf{x} = 0\}$, where $\mathcal{F}(\mathcal{X})$ denotes the space of smooth functions on $\mathcal{X}$. The cotangent space at $\rho$, $\mathcal{T}_\rho^* \mathcal{P}_2(\mathcal{X})$, is identified with the quotient space $\mathcal{F}(\mathcal{X})/\mathbb{R}$.
A metric tensor $G(\rho) : \mathcal{T}_\rho \mathcal{P}_2(\mathcal{X}) \to \mathcal{T}_\rho^* \mathcal{P}_2(\mathcal{X})$ is an invertible mapping defining an inner product on $\mathcal{T}_\rho \mathcal{P}_2(\mathcal{X})$:

$$g_\rho(\sigma_1, \sigma_2) = \int \sigma_1 G(\rho) \sigma_2 d\mathbf{x} = \int \Phi_1 G(\rho)^{-1} \Phi_2 d\mathbf{x},$$

where $\sigma_1, \sigma_2 \in \mathcal{T}_\rho \mathcal{P}_2(\mathcal{X})$ and $\Phi_i \in \mathcal{T}_\rho^* \mathcal{P}_2(\mathcal{X})$ is the function satisfying $\sigma_i = G(\rho)^{-1} \Phi_i$, for $i = 1, 2$.

**Definition 2.2** (Gradient Flow in Probability Spaces). The gradient flow of $F(\rho)$ under metric $G(\rho)$ is

$$\dot{\rho}_t = -G(\rho_t)^{-1} \delta_\rho F(\rho_t),$$

where $\delta_\rho F(\rho) : \mathcal{X} \to \mathbb{R}$ is the first variation of $F$, satisfying

$$F(\rho + \epsilon \sigma) = F(\rho) + \epsilon \int \delta_\rho F(\rho)(\mathbf{x}) \sigma(\mathbf{x}) d\mathbf{x},$$

for all $\sigma \in \mathcal{T}_\rho \mathcal{P}_2(\mathcal{X})$.

We focus on the Wasserstein-2 metric $\mathcal{W}_2$, widely used in practice (Nguyen & Tsuda, 2023; Nguyen et al., 2025b; Nguyen & Sakurai, 2023; 2024). The inverse Wasserstein metric tensor is given by

$$G^W(\rho)^{-1}\Phi = -\nabla \cdot (\rho \nabla \Phi),$$

where $\nabla \cdot$ denotes the divergence operator, and the metric is

$$g_\rho^W(\sigma_1, \sigma_2) = \int \sigma_1 G^W(\rho)\sigma_2 d\mathbf{x}$$
$$= \int \Phi_1 G^W(\rho)^{-1}\Phi_2 d\mathbf{x} = \int \langle \nabla \Phi_1, \nabla \Phi_2 \rangle d\rho.$$

Then, the resulting Wasserstein gradient flow is

$$\dot{\rho}_t = -\operatorname{grad} F(\rho_t) = -G^W(\rho_t)^{-1}\delta_\rho F(\rho_t)$$
$$= \nabla \cdot (\rho_t \nabla \delta_\rho F(\rho_t)),$$

where $\operatorname{grad} F(\rho) \in \mathcal{T}_\rho \mathcal{P}_2(\mathcal{X})$ is the Wasserstein gradient of $F$ at $\rho$. See (Wang & Li, 2022; Nguyen & Sakurai, 2023; 2024) for further details.

**Hamiltonian Flows in Probability Spaces.** Analogous to the Euclidean case, we can define the Hamiltonian

$$H(\rho, \Phi) = F(\rho) + \frac{1}{2}\int \Phi G(\rho)^{-1}\Phi d\mathbf{x},$$

where $\rho$ and $\Phi$ act as the state and momentum variables, respectively. Following the damped Hamiltonian interpretation of Nesterov's acceleration (Taghvaei & Mehta, 2019; Wang & Li, 2022), the Accelerated Information Gradient (AIG) flow is given by

$$\dot{\rho}_t = \delta_\Phi H(\rho_t, \Phi_t),$$
$$\dot{\Phi}_t + \alpha_t \Phi_t + \delta_\rho H(\rho_t, \Phi_t) = 0.$$

Under the Wasserstein-2 metric, this reduces to the Wasserstein accelerated gradient flow (W-AIG) (Wang & Li, 2022)

$$\dot{\rho}_t + \nabla \cdot (\rho_t \nabla \Phi_t) = 0,$$
$$\dot{\Phi}_t + \alpha_t \Phi_t + \frac{1}{2}\|\Phi_t\|_2^2 + \delta_\rho F(\rho_t) = 0. \tag{4}$$

## 2.3. Multiple Wasserstein Gradient *Descent* Algorithm for MODO

The goal of MODO (1) is to optimize multiple objectives over probability space simultaneously. As in classical MOO, different distributions may perform better on different objectives, which motivates the notion of Pareto optimality.

**Definition 2.3.** Consider the optimization problem (1).

1. A distribution $p^* \in \mathcal{P}_2(\mathcal{X})$ is Pareto optimal if there does not exist another $q \in \mathcal{P}_2(\mathcal{X})$ such that $F_k(q) \leq F_k(p^*)$ for all $k \in [K]$ and $F_l(q) < F_l(p^*)$ for at least one index $l \in [K]$. We denote the set of all Pareto optimal distributions by $P^*$.

2. A distribution $p^* \in \mathcal{P}_2(\mathcal{X})$ is weakly Pareto optimal if there does not exist another $q \in \mathcal{P}_2(\mathcal{X})$ such that $F_k(q) < F_k(p^*)$ for all $k \in [K]$. We denote the set of all weakly Pareto optimal distributions by $P_w^*$.

From Definition 2.3, it follows that $P^* \subseteq P_w^*$. However, the definition cannot directly be used in practice to check wheather a distribution is Pareto optimal. Nguyen et al. (2025a) introduced the definition of *Pareto stationary distribution*, which extends the definition of Pareto stationary points in Euclidean space, as follows.

**Definition 2.4.** We denote the probability simplex as Let $\mathcal{W} = \left\{ \mathbf{w} = (w_1, ..., w_K)^\top | \mathbf{w} \geq 0, \sum_{k=1}^K w_k = 1 \right\}$.

A distribution $\rho \in \mathcal{P}_2(\mathcal{X})$ is Pareto stationary or critical if

$$\min_{\mathbf{w} \in \mathcal{W}} \langle \operatorname{grad} \mathbf{F}(\rho)\mathbf{w}, \operatorname{grad} \mathbf{F}(\rho)\mathbf{w} \rangle_\rho = 0,$$

where
$\operatorname{grad} \mathbf{F}(\rho)(\mathbf{x}) = [\operatorname{grad} F_1(\rho)(\mathbf{x}), ..., \operatorname{grad} F_K(\rho)(\mathbf{x})]$, and $\mathbf{grad F}(\rho)(\mathbf{x})\mathbf{w} = \sum_{k=1}^K w_k \operatorname{grad} F_k(\rho)(\mathbf{x})$. We denote the set of all Pareto stationary distributions by $P_c^*$.

It can be verified that, in the geodesically convex setting, the stationarity conditions are also sufficient conditions for weak Pareto optimality, and we have that $P^* \subseteq P_w^* = P_c^*$.

MWGraD (Nguyen et al., 2025a) is designed to find a Pareto stationary distribution by iteratively constructing distributions $\rho_0, \rho_1, \ldots, \rho_T$, starting from a simple initial distribution $\rho_0$ (e.g., a standard Gaussian), and simultaneously decreasing all objectives . At iteration $n$, MWGraD seeks a tangent direction $s_n$ that maximizes the minimum decrease across objectives:

$$\max_{s \in T_{\rho_n}\mathcal{P}_2(\mathcal{X})} \min_{k \in [K]} \frac{1}{h}\left(F_k(\rho_n) - F_k(\gamma(h))\right)$$
$$\approx \max_{\mathbf{v} \in \mathcal{V}} \min_{k \in [K]} \int \langle \nabla \delta F_k(\rho_n), \mathbf{v} \rangle d\rho_n, \tag{5}$$

where $\mathbf{v} : \mathcal{X} \to \mathcal{X}$ is a vector field belonging to the space of vector fields $\mathcal{V}$; $s$ and $\mathbf{v}$ are related through the elliptic equation $s = \nabla \cdot (q_n \mathbf{v})$. Here, $\gamma : [0, 1] \to \mathcal{P}_2(\mathcal{X})$ is a curve satisfying that $\gamma(0) = \rho_n$ and $\gamma'(0) = s$. To regularize the update direction (i.e., vector field $\mathbf{v}$), a regularization term is introduced to (5) and we can solve for $s_n$ by optimizing

$$\max_{\mathbf{v} \in \mathcal{V}} \min_{k \in [K]} \int_{\mathcal{X}} \langle \nabla \delta_\rho F_k(\rho_n), \mathbf{v} \rangle d\rho_n - \frac{1}{2}\int_{\mathcal{X}} \|\mathbf{v}\|_2^2 d\rho_n. \tag{6}$$

The optimal $\mathbf{v}_n$ to (6) is

$$\mathbf{v}_n = \sum_{k=1}^K w_{n,k} \nabla \delta_\rho F_k(\rho_n), \tag{7}$$

where

$$\mathbf{w}_n = \underset{\mathbf{w} \in \mathcal{W}}{\arg\min} \frac{1}{2} \int \left\| \sum_{k=1}^{K} w_k \nabla \delta_\rho F_k(\rho_n) \right\|_2^2 d\rho_n. \quad (8)$$

The optimal $\mathbf{v}_n$ is used to update the current particles from $\mathbf{x}_n \sim \rho_n$ to $\mathbf{x}_{n+1} \sim \rho_{n+1}$ via $\mathbf{x}_{n+1} = \mathbf{x}_n - \eta_n \mathbf{v}_n(\mathbf{x}_n)$.

# 3. Multiple Wasserstein Gradient *Flow* and Its Acceleration

In this section, we formulate a continuous-time counterpart of MWGraD and study its theoretical theoretical properties, including convergence guarantees and an explicit characterization in the Gaussian family.

The velocity field $\mathbf{v}_n$ (7) specifies how particles $\mathbf{x}_n \sim \rho_n$ are transported to $\mathbf{x}_{n+1} \sim \rho_{n+1}$. Formally taking the limit $\eta_n \to 0$ yields the following probability flow, which we term the *MWGraD flow*:

$$\dot{\rho}_t + \nabla \cdot (\rho_t \nabla \Phi_t) = 0,$$
$$\Phi_t + \text{proj}_{\mathcal{C}(\rho_t), \rho_t}[0] = 0, \quad (9)$$

where $\mathcal{C}(\rho) = \text{conv}\left(\{\delta_\rho F_k(\rho) : k = 1, 2, \ldots, K\}\right)$ denotes the convex hull of the first variations of objectives $F_k$ ($k \in [K]$). Here, $\text{proj}_{\mathcal{K}, \rho}[f]$ denotes the projection of a function $f \in \mathcal{T}_\rho^* \mathcal{P}(\mathcal{X})$ onto a closed convex set $\mathcal{K} \subset \mathcal{T}_\rho^* \mathcal{P}(\mathcal{X})$ under the Wasserstein-2 metric:

$$\text{proj}_{\mathcal{K}, \rho}[f] = \underset{h \in \mathcal{K}}{\arg\min} \int (f - h) G^W(\rho)^{-1} (f - h) d\mathbf{x}$$
$$= \underset{h \in \mathcal{K}}{\arg\min} \int \|\nabla f - \nabla h\|_2^2 d\rho. \quad (10)$$

This formulation makes explicit the interpretation of MW-GraD as a continuous-time gradient flow on probability space, where convex combinations of Wasserstein gradients induce a Pareto-stationary descent direction. Note that the second equation in the MWGraD flow is defined only up to an additive function of $t$, which does not affect the dynamics, since only $\nabla \Phi_t$ appears in the continuity equation.

We next characterize the MWGraD flow explicitly in the Gaussian family when the objective functions are KL divergences. In this case, the infinite-dimensional flow (9) reduces to an ODE governing the evolution of the covariance matrix.

**Theorem 3.1** (MWGraD flow in Gaussian family). *Suppose the initial distribution $\rho_0$ and the target distributions $\pi_k$, for $k \in [K]$, are zero-mean Gaussian distributions with covariance matrices $\Sigma_0$ and $\Sigma_*^k$, respectively. Let $\rho$ be a zero-mean Gaussian with covariance matrix $\Sigma$, and let the*

*$k$-th objective function be the KL divergence from $\rho$ to $\pi_k$. Define*

$$F_k^\dagger(\Sigma) := F_k(\rho) = \text{KL}(\rho \| \pi_k) =$$
$$= \frac{1}{2} \left[ \text{tr}(\Sigma(\Sigma_*^k)^{-1}) - \log\det\left(\Sigma(\Sigma_*^k)^{-1}\right) - d \right].$$

*Let $\Sigma_t$ satisfy*

$$\dot{\Sigma}_t - 2(S_t \Sigma_t + \Sigma_t S_t) = 0,$$
$$S_t = -\sum_{k=1}^{K} w_{t,k} \nabla_{\Sigma_t} F_k^\dagger(\Sigma_t), \quad (11)$$

*where $\mathbf{w}_t = \underset{\mathbf{w} \in \mathcal{W}}{\arg\min} \left\| \sum_{k=1}^{K} w_k \nabla_{\Sigma_t} F_k^\dagger(\Sigma_t) \right\|_{\Sigma_t}^2$, with initial condition $\Sigma_0 \succ 0$. Then $\Sigma_t \succ 0$ for any $t \geq 0$ (i.e., positive definite). Furthermore, defining $\rho_t = \mathcal{N}(0, \Sigma_t)$ and $\Phi_t(\mathbf{x}) = \mathbf{x}^\top S_t \mathbf{x}$, the pair $(\rho_t, \Phi_t)$ is a solution to the MWGraD flow (9) with initial condition $\rho_0$ and $\Phi_0 = 0$.*

The proof is provided in Appendix D. Theorem 3.1 provides an explicit characterization of the MWGraD flow within the Gaussian family, reducing the infinite-dimensional evolution in probability space to a finite-dimensional matrix-valued ODE. Establishing existence, uniqueness and well-posedness of the MWGraD flow in more general distributional settings remains an interesting direction of future work.

### 3.1. Convergence Analysis of MWGraD flow

To characterize convergence of the MWGraD flow (9) for MODO (1), we introduce the following *merit function*:

$$\mathcal{M}(\rho) = \sup_{q \in \mathcal{P}_2(\mathcal{X})} \min_{k \in [K]} \{F_k(\rho) - F_k(q)\}. \quad (12)$$

This definition naturally generalizes the merit function used to analyze convergence of function values for MOO in Euclidean spaces (Sonntag & Peitz, 2024; Tanabe et al., 2023). Importantly, for all $\rho \in \mathcal{P}_2(\mathcal{X})$, we have $\mathcal{M}(\rho) \geq 0$, and $\rho$ is weakly Pareto optimal if and only if $\mathcal{M}(\rho) = 0$. This follows directly from Theorem 3.1 of (Tanabe et al., 2023).

We establish the convergence rate of the MWGraD flow (9) by showing that $\mathcal{M}(\rho_t)$ converges to zero under the following assumption.

**Assumption 3.2.** For $\mathbf{u} \in \mathbb{R}^K$, define the level set: $\Omega_{\mathbf{F}}(\mathbf{u}) = \{q \in \mathcal{P}_2(\mathcal{X}) | \mathbf{F}(q) \leq \mathbf{u}\}$, where $\leq$ denotes the componentwise order in $\mathbb{R}^K$. Assume that for every $q \in \mathcal{P}_2(\mathcal{X})$, there exists $p^* \in P_w^*$ such that $\mathbf{F}(p^*) \leq \mathbf{F}(q)$, and that

$$R = \sup_{\mathbf{F}^* \in \mathbf{F}(P_w^* \cap \Omega_{\mathbf{F}}(\mathbf{F}(\rho_0)))} \inf_{q \in \mathbf{F}^{-1}(\{\mathbf{F}^*\})} \mathcal{W}_2^2(q, \rho_0) < \infty.$$

**Remarks.**

- Assumption 3.2 is extended from Assumption 5.2 in (Tanabe et al., 2023) from the Euclidean space to probability space.

- In the single-objective case ($K = 1$), Assumption 3.2 holds if the optimization problem has at least one optimal solution. Indeed, for $K = 1$, $P_w^*$ coincides with the set of optimal solutions, $P_w^* \cap \Omega_{\mathbf{F}}(\mathbf{F}(p_0)) = P_w^*$, and $R = \inf_{p^* \in P_w^*} \mathcal{W}_2^2(p^*, p_0) < \infty$.

To facilitate the convergence analysis, we next introduce an auxiliary lemma that allows evaluating the merit function (12) without taking the supremum over the entire space $\mathcal{P}_2(\mathcal{X})$.

**Lemma 3.3.** *Let $\rho_0 \in \mathcal{P}_2(\mathcal{X})$ and $\rho \in \Omega_{\mathbf{F}}(\mathbf{F}(\rho_0))$. For $k \in [K]$, define*

$$E_k(\rho, q) = F_k(\rho) - F_k(q), E(\rho, q) = \min_{k \in [K]} E_k(\rho, q).$$

*Then*

$$\sup_{\mathbf{F}^* \in \mathbf{F}(P_w^* \cap \Omega_{\mathbf{F}}(\mathbf{F}(\rho_0)))} \inf_{q \in \mathbf{F}^{-1}(\mathbf{F}^*)} E(\rho, q) = \sup_{q \in \mathcal{P}_2(\mathcal{X})} E(\rho, q).$$

The proof is deferred to Appendix A. We further impose the following assumption on the objectives $F_k$ (for $k \in [K]$).

**Assumption 3.4** ($\beta$-strong geodesic convexity and regularity)**.** For each $k \in [K]$, assume that $F_k : \mathcal{P}_2(\mathcal{X}) \to \mathbb{R}$ is $\beta$-strongly geodesically convex with respect to the Wasserstein-2 distance. Moreover, for every $p, q \in \mathcal{P}_2(\mathcal{X})$ considered in the analysis, assume that the optimal transport map $T$ from $p$ to $q$ exists. Then

$$F_k(q) \geq F_k(p) + \langle \text{grad } F_k(p), \text{Exp}_p^{-1}(q) \rangle_p + \frac{\beta}{2} \mathcal{W}_2^2(p, q)$$

$$= F_k(p) + \int \langle \nabla \delta_p F_k(p)(\mathbf{x}), T(\mathbf{x}) - \mathbf{x} \rangle dp(\mathbf{x})$$

$$+ \frac{\beta}{2} \int \|T(\mathbf{x}) - \mathbf{x}\|_2^2 dp(\mathbf{x}),$$

where $\text{Exp}_p$ denotes the exponential mapping, which specifies how to move $p$ along a tangent vector on $\mathcal{P}_2(\mathcal{X})$, $\text{Exp}_p^{-1}$ denotes its inversion mapping, which maps a point on $\mathcal{P}_2(\mathcal{X})$ to a tangent vector. If $\beta = 0$, $F_k$ is said to be geodesically convex with respect to the Wasserstein-2 distance. We refer to (Santambrogio, 2015) for details.

We are now ready to state the convergence result for the MWGraD flow (9).

**Theorem 3.5.** *Let $\rho_t$ be the solution of the MWGraD flow (9). Suppose that each $F_k$ is geodesically convex for $k \in [K]$, and that Assumption 3.2 holds. Assume additionally that, for every $q \in \mathcal{P}_2(\mathcal{X})$, the optimal transport map $T_t$ from $\rho_t$ to $q$ exists and is differentiable. Then for all $t \geq 0$,*

$$\mathcal{M}(\rho_t) \leq \frac{R}{2t} = \mathcal{O}\left(\frac{1}{t}\right).$$

*Sketch of Proof.* (See Appendix B for details) For any $q \in \mathcal{P}_2(\mathcal{X})$, define the Lyapunov functional

$$\mathcal{V}(q, t) = \frac{1}{2} \mathcal{W}_2^2(\rho_t, q) = \frac{1}{2} \int \|T_t(\mathbf{x}) - \mathbf{x}\|_2^2 d\rho_t(\mathbf{x}),$$

where $T_t$ is the optimal transport map from $\rho_t$ to $q$. We can show that $E_k(\rho_t, q)$, for $k \in [K]$, and $E(\rho_t, q)$ are non-increasing in time, i.e., $\dot{E}_k(\rho_t, q) \leq 0$ for all $k$. Moreover,

$$E(\rho_t, q) \leq -\dot{\mathcal{V}}(q, t). \tag{13}$$

Integrating (13) over $[0, t]$ gives

$$E(\rho_t, q) \leq \frac{\mathcal{V}(q, 0)}{t} = \frac{\mathcal{W}_2^2(\rho_0, q)}{2t}.$$

By Assumption 3.2,

$$\sup_{\mathbf{F}^* \in \mathbf{F}(P^* \cap \Omega_{\mathbf{F}}(\mathbf{F}(p_0)))} \inf_{q \in \mathbf{F}^{-1}(\{\mathbf{F}^*\})} E(\rho_t, q) \leq \frac{R}{2t}.$$

Since $E_k(\rho_t, q)$ is non-increasing, it follows that $F_k(\rho_t) - F_k(q) \leq F_k(\rho_0) - F_k(q)$ for $k \in [K]$, which means that $\rho_t \in \Omega_{\mathbf{F}}(\mathbf{F}(\rho_0))$. Applying Lemma 3.3 completes the proof.

### 3.2. Accelerated MWGraD Flow and Its Convergence Analysis

Inspired by AIG flow (Taghvaei & Mehta, 2019; Wang & Li, 2022), we introduce the *accelerated MWGraD flow* (A-MWGraD) as follows:

$$\dot{\rho}_t + \nabla \cdot (\rho_t \nabla \Phi_t) = 0,$$
$$\dot{\Phi}_t + \alpha_t \Phi_t + \frac{1}{2} \|\nabla \Phi_t\|_2^2 + \text{proj}_{\mathcal{C}(\rho_t), \rho_t}[0] = 0. \tag{14}$$

As in the MWGraD flow, the second equation of the A-MWGraD flow (14) is defined only up to an additive function of $t$, which does not affect the dynamics. Furthermore, when $K = 1$, (14) reduces to the Wasserstein accelerated information gradient (W-AIG) flow (4).

We next provide an explicit characterization of the A-MWGraD flow in the Gaussian family when the objective functionals are KL divergences.

**Theorem 3.6** (A-MWGraD flow in Gaussian family)**.** *Suppose the initial distribution $\rho_0$ and the target distributions $\pi_k$, for $k \in [K]$, are zero-mean Gaussian distributions with covariance matrices $\Sigma_0$ and $\Sigma_*^k$, respectively. Let $(\Sigma_t, S_t)$ satisfy*

$$\dot{\Sigma}_t - 2(S_t \Sigma_t + \Sigma_t S_t) = 0,$$
$$\dot{S}_t + \alpha_t S_t + 2S_t^2 + \sum_{k=1}^{K} w_{t,k} \nabla_{\Sigma_t} F_k^\dagger(\Sigma_t) = 0, \tag{15}$$

*where* $\mathbf{w}_t = \arg\min_{\mathbf{w}\in\mathcal{W}} \left\| \sum_{k=1}^{K} w_k \nabla_{\Sigma_t} F_k^{\dagger}(\Sigma_t) \right\|_{\Sigma_t}^2$, *with initial condition* $\Sigma_0 \succ 0$. *Then* $\Sigma_t \succ 0$ *for any* $t \geq 0$. *Furthermore, defining* $\rho_t = \mathcal{N}(0, \Sigma_t)$ *and* $\Phi_t(\mathbf{x}) = \mathbf{x}^\top S_t \mathbf{x} + C(t)$, *where*

$$C(t) = -t + \frac{1}{2}\sum_{k=1}^{K}\int_0^t w_{s,k}\log\det\left(\Sigma_s\left(\Sigma_*^k\right)^{-1}\right)ds.$$

*Then, the pair* $(\rho_t, \Phi_t)$ *is a solution to the A-MWGraD flow (14) with initial condition* $\rho_0$ *and* $\Phi_0 = 0$.

The proof is provided in Appendix D. The theorem gives an explicit construction of solutions to the A-MWGraD flow when the distributions are restricted to the Gaussian family and the objective functions are KL divergences.

We next establish convergence rates for the A-MWGraD flow. Specifically, we show that the merit function $\mathcal{M}(\rho_t)$ converges to zero at the accelerated rate $\mathcal{O}(t^{-2})$ when the objectives are geodesically convex, and at the exponential rate $\mathcal{O}(e^{-\sqrt{\beta}t})$ when the objectives are $\beta$-strongly geodesically convex.

**Theorem 3.7.** *Let* $\rho_t$ *be the solution of the A-MWGraD flow (14). Suppose that Assumption 3.2 holds. Assume additionally that, for every* $q \in \mathcal{P}_2(\mathcal{X})$, *the optimal transport map* $T_t$ *from* $\rho_t$ *to* $q$ *exists and is differentiable.*

*First, assume that each* $F_k$ *is geodesically convex for* $k \in [K]$, *and let* $\alpha_t = \alpha/t$ *with* $\alpha \geq 3$. *Then, for all* $t \geq 0$,

$$\mathcal{M}(\rho_t) \leq \frac{(\alpha-1)R}{t^2} = \mathcal{O}\left(\frac{1}{t^2}\right). \quad (16)$$

*Second, assume that each* $F_k$ *is* $\beta$-*strongly geodesically convex for* $k \in [K]$ *with* $\beta > 0$, *and let* $\alpha_t = 2\sqrt{\beta}$. *Then, for all* $t \geq 0$,

$$\mathcal{M}(\rho_t) \leq e^{-\sqrt{\beta}t}\left[\min_{l\in[K]}F_l(\rho_0) + \frac{\beta R}{2}\right] = \mathcal{O}(e^{-\sqrt{\beta}t}). \quad (17)$$

*Sketch of Proof of (16).* (See Appendix C for the detailed proofs of (16) and (17)) For each $k \in [K]$, define the Lyapunov functional

$$\mathcal{E}_{k,\lambda}(\rho_t, q) = t^2(F_k(\rho_t) - F_k(q))$$
$$+ \frac{1}{2}\int \|2(\mathbf{x} - T_t(\mathbf{x})) + t\nabla\Phi_t(\mathbf{x})\|_2^2 d\rho_t(\mathbf{x})$$
$$+ \frac{\lambda}{2}\int \|\mathbf{x} - T_t(\mathbf{x})\|_2^2 d\rho_t(\mathbf{x}),$$

and let

$$\mathcal{E}_{\lambda}(\rho_t, q) = \min_{k\in[K]}\mathcal{E}_{k,\lambda}(\rho_t, q).$$

With the choice $\lambda = 2(\alpha-3)$, it can be shown that $\mathcal{E}_{\lambda}(\rho_t, q)$ is non-increasing in time, i.e. $\dot{\mathcal{E}}_{\lambda}(\rho_t, q) \leq 0$. Consequently,

$$E(\rho_t, q) = \min_{k\in[K]}\{F_k(\rho_t) - F_k(q)\} \leq \frac{\mathcal{E}_{\lambda}(\rho_t, q)}{t^2}$$
$$\leq \frac{\mathcal{E}_{\lambda}(\rho_0, q)}{t^2} = \frac{(\alpha-1)\mathcal{W}_2^2(\rho_0, q)}{t^2}.$$

By Assumption 3.2, this yields

$$\sup_{\mathbf{F}^*\in\mathbf{F}(P_w^*\cap\Omega_{\mathbf{F}}(\mathbf{F}(p_0)))}\inf_{q\in\mathbf{F}^{-1}(\{\mathbf{F}^*\})}E(\rho_t, q) \leq \frac{(\alpha-1)R}{t^2}.$$

Moreover, by Corollary C.2, $F_k(\rho_t) \leq F_k(\rho_0)$ for all $k \in [K]$, which ensures that $\rho_t \in \Omega_{\mathbf{F}}(\mathbf{F}(\rho_0))$. Applying Lemma 3.3 completes the proof of (16).

### 3.3. Particle Implementation of A-MWGraD Flow

To design fast sampling algorithms, we reformulate the evolution of probability distributions in terms of particle dynamics. Suppose that $\mathbf{x}_t \sim \rho_t$ and $\mathbf{v}_t = \nabla\Phi_t(\mathbf{x}_t)$ denotes the position and the velocity of a particle at time $t$, respectively. Then the A-MWGraD flow (14) induces the following particle dynamics:

$$\dot{\mathbf{x}}_t = \mathbf{v}_t,$$
$$\dot{\mathbf{v}}_t + \alpha_t\mathbf{v}_t + \sum_{k=1}^{K}w_{t,k}\nabla\delta_\rho F_k(\rho_t)(\mathbf{x}_t) = 0. \quad (18)$$

See Appendix E for the detailed derivation.

A typical choice of $F_k(\rho)$ for sampling is the KL divergence

$$F_k(\rho) = KL(\rho\|\pi_k) = \int f_k d\rho + \int \log\rho \, d\rho,$$

where the target distributions satisfy $\pi_k \propto \exp(-f_k)$ for $k \in [K]$. In this case, the particle dynamics become

$$\dot{\mathbf{x}}_t = \mathbf{v}_t, \dot{\mathbf{v}}_t + \alpha_t\mathbf{v}_t + \sum_{k=1}^{K}w_{t,k}\Delta_k^{(t)}(\mathbf{x}_t) = 0,$$

where $\Delta_k^{(t)} = \nabla f_k + \nabla\log\rho_t$ is the Wasserstein gradient of the KL divergence.

**Discrete-Time Particle System.** Consider a particle systems $\{\mathbf{x}_i^{(0)}\}_{i=1}^m$ with initial velocities $\mathbf{v}_i^{(0)} = 0$ for $i \in [m]$. At iteration $n$, the discretized update with step size $\eta > 0$ is given by

$$\mathbf{x}_i^{(n+1)} = \mathbf{x}_i^{(n)} + \sqrt{\eta}\mathbf{v}_i^{(n)},$$
$$\mathbf{v}_i^{(n+1)} = \alpha_n\mathbf{v}_i^{(n)} - \sqrt{\eta}\sum_{k=1}^{K}w_{n,k}\Delta_k^{(n)}(\mathbf{x}_i^{(n)}), \quad (19)$$

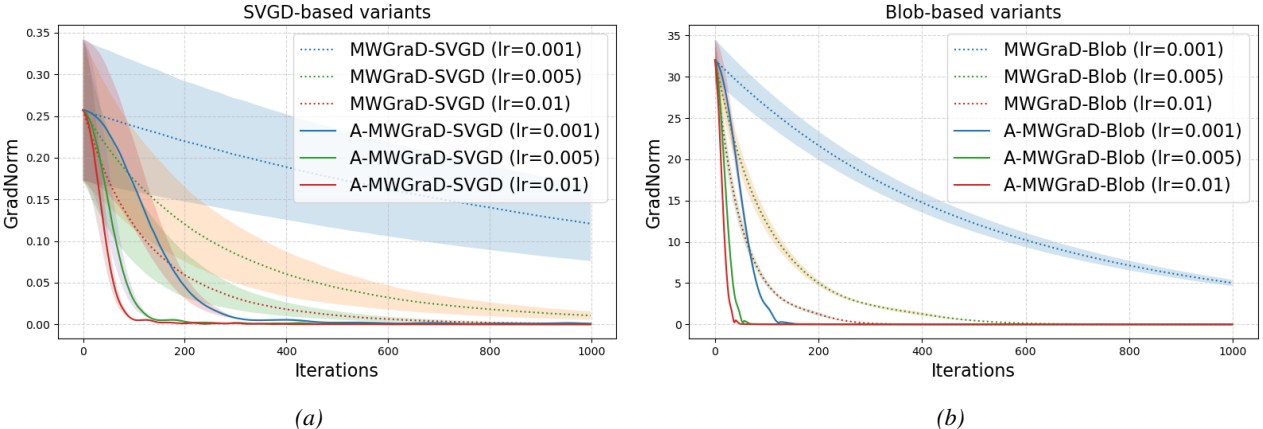

*Figure 1.* Convergence comparison: (a) SVGD-based variants, (b) Blob-based variants. The plot show mean and standard deviation of GradNorm over 1000 iterations with three different step sizes of $\eta = 0.001, 0.005, 0.01$.

where the momentum parameter is $\alpha_n = (n - 1)/(n + 2)$ which corresponds to the continuous-time choice $\alpha_t = 3/t$ when the objectives are geodesically convex, or $\alpha_n = (1 - \sqrt{\beta\eta})/(1 + \sqrt{\beta\eta})$ which corresponds to the continuous-time choice $\alpha_t = 2\sqrt{\beta}$ when the objectives are $\beta$-strongly geodesically convex. However, direct computation of $\log \rho_n(\mathbf{x})$ is not feasible for empirical measures $\rho_n(\mathbf{x})$. Following (Nguyen et al., 2025a), we therefore adopt kernel-based approximations, namely Stein Variational Gradient Descent (SVGD) (Liu & Wang, 2016) and Blob methods (Carrillo et al., 2019).

For SVGD (Liu & Wang, 2016), we approximate $\Delta_k^{(n)}$ vy $\bar{\Delta}_k^{(n)}$ (for $k \in [K]$), given by

$$\bar{\Delta}_k^{(n)}(\mathbf{x}) = \mathbb{E}_{\mathbf{y}\sim\rho_n}[K(\mathbf{x},\mathbf{y})(\nabla f_k(\mathbf{y}) + \nabla \log \rho_n(\mathbf{y}))]$$
$$= \int_{\mathcal{X}} K(\mathbf{x},\mathbf{y})\nabla f_k(\mathbf{y})\rho_n(\mathbf{y})d\mathbf{y} - \int_{\mathcal{X}} \nabla_{\mathbf{y}}K(\mathbf{x},\mathbf{y})\rho_n(\mathbf{y})d\mathbf{y},$$

where the second equality follows from integration by parts. Consequently, the particle approximation of $\Delta_k^{(n)}$ is

$$\bar{\Delta}_k^{(n)}(\mathbf{x}_i^{(n)}) = \sum_{j=1}^{m} K(\mathbf{x}_i^{(n)},\mathbf{x}_j^{(n)})\nabla f_k(\mathbf{x}_j^{(n)})$$
$$- \sum_{j=1}^{m} \nabla_{\mathbf{x}_j^{(n)}}K(\mathbf{x}_i^{(n)},\mathbf{x}_j^{(n)}). \quad (20)$$

For Blob methods (Carrillo et al., 2019), we approximate $\Delta_k^{(n)}$ as follows

$$\bar{\Delta}_k^{(n)}(\mathbf{x}_i^{(n)}) = \nabla f_k(\mathbf{x}_i^{(n)})$$
$$- \sum_{j=1}^{m} \nabla_{\mathbf{x}_j^{(n)}}K(\mathbf{x}_i^{(n)},\mathbf{x}_j^{(n)}) / \sum_{l=1}^{m} K(\mathbf{x}_j^{(n)},\mathbf{x}_l^{(n)})$$
$$- \sum_{j=1}^{m} \nabla_{\mathbf{x}_j^{(n)}}K(\mathbf{x}_i^{(n)},\mathbf{x}_j^{(n)}) / \sum_{l=1}^{m} K(\mathbf{x}_i^{(n)},\mathbf{x}_j^{(n)}). \quad (21)$$

We refer the reader to Proposition 3.12 of (Carrillo et al., 2019) for the detailed derivation. The overall algorithms of MWGraD and A-MWGraD are summarized in Appendix F.

The weights $\mathbf{w}_n$ in (19) can be approximated by minimizing the squared norm of the combined Wasserstein descent direction:

$$\mathbf{w}_n = \arg\min_{\mathbf{w}\in\mathcal{W}} \frac{1}{m}\sum_{i=1}^{m}\left\|\sum_{k=1}^{K} w_k\bar{\Delta}_k^{(n)}(\mathbf{x}_i^{(n)})\right\|_2^2. \quad (22)$$

## 4. Numerical Experiments

In this section, we present numerical experiments demonstrating the acceleration effect of A-MWGraD flow (14).

### 4.1. Experiments on toy examples

We consider the task of sampling from multiple target distributions, where each target is a mixture of two Gaussians: $\pi_k(\mathbf{x}) = \gamma_{k1}\mathcal{N}(\mathbf{x}|\mu_{k1},\Sigma_{k1}) + \gamma_{k2}\mathcal{N}(\mathbf{x}|\mu_{k2},\Sigma_{k2})$, $k = 1, 2, 3, 4$, with mixture weights $\gamma_{k1} = 0.7, \gamma_{k2} = 0.3$ for all $k$. The component means are given by $\mu_{11} = [4, -4]^\top$, $\mu_{12} = [0.1, 0.2]^\top$, $\mu_{21} = [-4, 4]^\top$, $\mu_{22} = [-0.1, 0.3]^\top$, $\mu_{31} = [-4, -4]^\top$, $\mu_{32} = [0.4, -0.4]^\top$, $\mu_{41} = [4, 4]^\top$, $\mu_{42} = [-0.2, 0.3]^\top$, and all covariance matrices $\Sigma_{kj}$ are set to the identity matrix of size $2 \times 2$, for $k = 1, 2, 3, 4$ and $j = 1, 2$.

We represent $\rho$ using 50 particles, initially sampled from a standard Gaussian. The particles are updated using MOO-SVGD (Liu et al., 2021), MWGraD variants (Nguyen et al., 2025a) (MWGraD-SVGD and MWGraD-Blob), and the proposed accelerated variants (A-MWGraD-SVGD and A-MWGraD-Blob). Figure 4 in Appendix F shows the target distributions sharing a common high-density region around the origin. To approximate $\bar{\Delta}_k^{(n)}(\mathbf{x})$ (see (20) and (21)), we use a Gaussian kernel with fixed bandwidth 1. For the

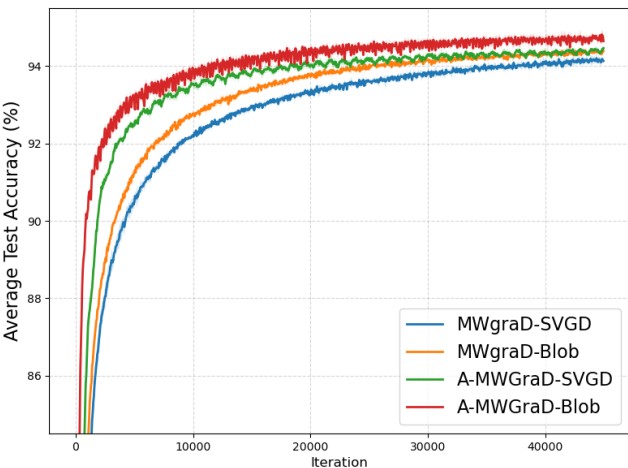

*(a)* Multi-MNIST

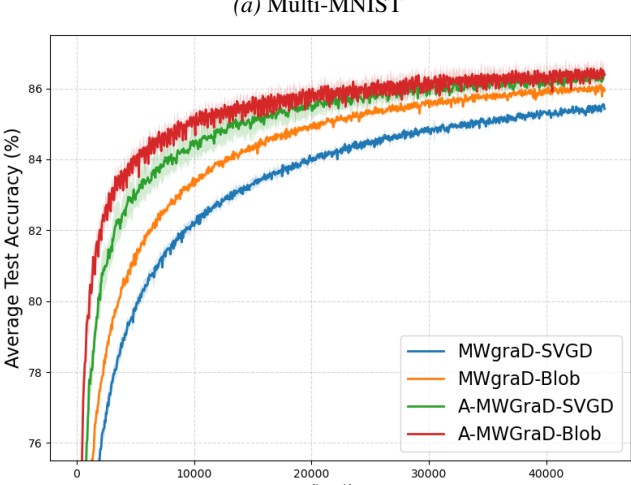

*(b)* Multi-Fashion

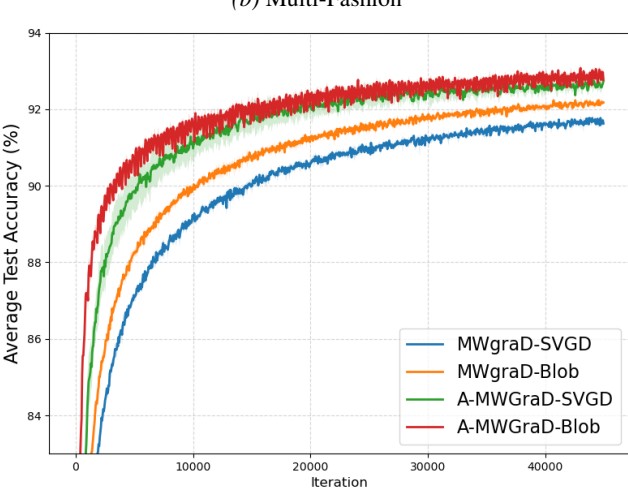

*(c)* Multi-Fashion-MNIST

*Figure 2.* Evaluation of the average test accuracies over 40000 training iterations on datasets: (a) Multi-MNIST, (b) Multi-Fashion, and (c) Multi-Fashion-MNIST.

A-MWGraD variants, we set $\alpha_n = (n-1)/(n+2)$. To compare MWGraD and A-MWGraD variants, we compute the approximate squared norm of the combined Wasserstein gradients over the particles:

$$\text{GradNorm} = \frac{1}{m}\sum_{i=1}^{m}\left\|\sum_{k=1}^{K}w_{n,k}\bar{\Delta}_{k}^{(n)}(\mathbf{x}_{i}^{(n)})\right\|_{2}^{2}, \quad (23)$$

where $\mathbf{w}_n$ is obtained by solving (22). It can be verified that (23) provides an approximation of the squared norm of the velocity $\mathbf{v}^{(n)}$ in (7). As (23) approaches zero, the distribution $\rho_n$ approaches a Pareto stationary distribution. In this experiment, we perform five independent trials and record the values of GradNorm. Figure 1 report the mean and standard deviation of this quantity over 1000 iterations of particle updates for both the SVGD-based and Blob-based variants. We evaluate three different step sizes of $\eta = 0.001, 0.005, 0.01$. We observe that, across all step sizes, the A-MWGraD variants consistently achieve low values of GradNorm faster than their MWGraD counterparts.

Furthermore, Figure 4 in Appendix F visualizes the evolution of the 50 particles updated by different methods across iterations using a fixed step size 0.001. We see that particles by MOO-SVGD spread across all modes of the target distributions after 2000 iterations, while particles updated by MWGraD and A-MWGraD variants concentrate on the shared high-density region. Notably, the A-MWGraD variants converge significantly faster than MW-GraD variants: for example, MWGraD-Blob requires approximately 500 iterations to cover the high-density region, while A-MWGraD-Blob achieves this within only 50 iterations. These results visually and quantitatively confirms the acceleration effect of A-MWGraD in this toy setting.

### 4.2. Experiments on Bayesian multi-task learning

We follow the experimental setup of (Phan et al., 2022; Nguyen et al., 2025a) to evaluate the acceleration effect of A-MWGraD on real-world datasets.

**Bayesian Multi-task Learning**. We consider a Bayesian multi-task learning problem with $K$ prediction tasks and training dataset $\mathbb{D}$. For each task $k \in [K]$, the model parameters are denoted by $\theta^k = [\mathbf{x}, \mathbf{z}^k]$, where $\mathbf{x}$ represent the shared component across tasks and $\mathbf{z}^k$ represents the task-specific component. Following (Phan et al., 2022), we maintain a set of $m$ models (particles) $\theta_i = [\theta_i^k]_{k=1}^K$ for $i \in [m]$, where $\theta_i^k = [\mathbf{x}_i, \mathbf{z}_i^k]$. At each iteration, given the task-specific components $\mathbf{z}_i^k$ for $i \in [m], k \in [K]$, we sample the shared component from the multiple target posteriors $p(\mathbf{x}|\mathbf{z}^k, \mathbb{D})$, $k \in [K]$ using either MWGraD or A-MWGraD variants. Then, for each task $k$, the task-specific components $[\mathbf{z}_i^k]_{i=1}^m$ are updated by sampling from the posterior $p(\mathbf{z}^k|\mathbf{x}, \mathbb{D})$. This procedure corresponds to standard

| Datasets | Tasks | MOO -SVGD | MWGraD -SVGD | MWGraD -Blob | A-MWGraD -SVGD | A-MWGraD -Blob |
|---|---|---|---|---|---|---|
| Multi-Fashion +MNIST | #1 | 94.8±0.4 | 94.7±0.3 | 94.1±0.5 | **96.4**±0.4 | 96.1±0.5 |
| | #2 | 85.6±0.2 | 88.9±0.6 | **90.5**±0.4 | 90.3±0.3 | **90.7**±0.4 |
| Multi-MNIST | #1 | 93.1±0.3 | 95.3±0.7 | 94.9±0.2 | 95.3±0.5 | **95.6**±0.4 |
| | #2 | 91.2±0.2 | 92.9±0.5 | 93.6±0.5 | 93.4±0.4 | **94.2**±0.4 |
| Multi-Fashion | #1 | 83.8±0.8 | 85.9±0.6 | 85.8±0.3 | 85.1±0.4 | **86.3**±0.5 |
| | #2 | 83.1±0.3 | 85.6±0.5 | 86.3±0.5 | **87.4**±0.6 | 86.5±0.7 |

*Table 1.* Experimental results on Multi-Fashion+MNIST, Multi-MNIST, and Multi-Fashion. Ensemble accuracy (higher is better) averaged over three independent runs with different initializations.

Bayesian particle-based inference and can be implemented using SVGD, Blob, or neural-network samplers (Nguyen et al., 2025a; Phan et al., 2022). For sampling the shared component, we apply the proposed accelerated variants A-MWGraD-SVGD and A-MWGraD-Blob, and consider the following baselines:

- MGDA (Désidéri, 2012), a classical multi-objective optimization method in Euclidean space applied to Bayesian multi-task learning by training separable models.

- MOO-SVGD (Liu et al., 2021), which incorporates multi-objective optimization into the SVGD framework to encourage particle diversity across targets.

- MT-SGD (Phan et al., 2022), an SVGD-based multi-target sampling method designed to generate diverse particles in high-density regions shared across targets.

- MWGraD variants (Nguyen et al., 2025a), which perform multi-objective optimization directly in probability spaces without acceleration.

In our implementation, MWGraD with the SVGD gradient approximation coincides with MT-SGD; therefore we report results only for MWGraD-SVGD. Furthermore, previous studies (Nguyen et al., 2025a) show that MOO-SVGD and MWGraD variants consistently outperform MGDA in this setting, so MGDA is omitted from the comparison.

**Datasets and Evaluation Metric**. We evaluate the methods on three benchmark datasets: Multi-Fashion-MNIST (Sabour et al., 2017), Multi-MNIST, and Multi-Fashion (Phan et al., 2022). Each dataset contains 120,000 training and 20,000 testing images generated by overlaying samples from MNIST and FashionMNIST, resulting in a two-task classification problem where each image is associated with two labels. We compare the proposed methods against MOO-SVGD and MWGraD variants. All results are reported using ensemble predictions from five particle models.

**Results**. Following the toy example, we evaluate the acceleration effect of A-MWGraD by comparing MWGraD variants (MWGraD-SVGD and MWGraD-Blob) with their accelerated counterparts (A-MWGraD-SVGD and A-MWGraD-Blob) in terms of the average test accuracy over training iterations. Figure 2 reports the mean accuracy (of two task-wise accuracies) averaged over five independent trials in 40000 iterations for the three considered datasets. We observe that, on most datasets, A-MWGraD variants reach higher test accuracy faster than MWGraD, demonstrating the acceleration effect of A-MWGraD for these tasks. Table 1 reports the test ensemble accuracy of compared methods for each task after 40000 iterations. We observe that variants of A-MWGraD often improve convergence compared to unaccelerated ones while achieving competitive or superior performance in several experiments. For instance, A-MWGraD-SVGD and A-MWGraD-Blob achieve the best performance of 96.4% and 90.2% for task 1 and 2, repsectively, on Multi-Fashion+MNIST. On Multi-MNIST, A-MWGraD-Blob achieves the best performance on both tasks (95.6% and 94.2%). These results show that A-MWGraD variants outperform MWGraD variants not only in convergence speed but also sampling effectiveness on multi-target sampling tasks.

## 5. Conclusions

We studied multi-objective optimization over probability spaces (MODO) and focused on the MWGraD algorithm (Nguyen et al., 2025b), designed to solve this problem. We proposed A-MWGraD, an accelerated variant of MWGraD, based on damped Hamiltonian dynamics underlying Nesterov's acceleration (Nesterov, 2013). We further conducted both theoretical and empirical analyses to demonstrate its acceleration effect relative to the original method. Despite these advances, several limitations remain: (i) the discrete-time convergence rates of A-MWGraD remain unestablished, and (ii) our convergence analysis assumes exact computation of Wasserstein gradients of all objectives, which is generally infeasible in practice. Our future work will address these issues and explore broader applications of MODO and A-MWGraD beyond multi-task learning.

## Acknowledgements

This work is supported by International Collaborative Research Program of Institute for Chemical Research of Kyoto University [grant number: #2026-20], MEXT KAKENHI [grant number: 26K21294, 25H01144, 26K21756, 24H00685], ICR's iJURC Joint Research Projects in 2025 and 2026.

## Impact Statement

This paper presents work whose goal is to advance the field of Bayesian Machine Learning. There are many potential social consequences to collecting user data, even privately, none of which we feel must be specified here.

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

## A. Detailed Proof of Lemma 3.3

*Proof.* Recall that

$$E(\rho, q) = \min_{k \in [K]} \{F_k(\rho) - F_k(q)\}.$$

Then

$$\sup_{\mathbf{F}^* \in \mathbf{F}(P_w^* \cap \Omega_\mathbf{F}(\mathbf{F}(\rho_0)))} \inf_{q \in \mathbf{F}^{-1}(\mathbf{F}^*)} E(\rho, q) = \sup_{\mathbf{F}^* \in \mathbf{F}(P_w^* \cap \Omega_\mathbf{F}(\mathbf{F}(\rho_0)))} \min_{k \in [K]} \{F_k(\rho) - F_k^*\}$$

$$= \sup_{q \in P_w^* \cap \Omega_\mathbf{F}(\mathbf{F}(\rho_0))} \min_{k \in [K]} \{F_k(\rho) - F_k(q)\}.$$

Since $\rho \in \Omega_\mathbf{F}(\mathbf{F}(\rho_0))$, we have $F_k(\rho) \le F_k(\rho_0)$ for $k \in [K]$. Hence,

$$\sup_{q \in P_w^* \cap \Omega_\mathbf{F}(\mathbf{F}(\rho_0))} \min_{k \in [K]} \{F_k(\rho) - F_k(q)\} = \sup_{q \in \Omega_\mathbf{F}(\mathbf{F}(\rho_0))} \min_{k \in [K]} \{F_k(\rho) - F_k(q)\}$$

$$= \sup_{q \in \mathcal{P}_2(\mathcal{X})} \min_{k \in [K]} \{F_k(\rho) - F_k(q)\},$$

which completes the proof. □

## B. Detailed Proof of Theorem 3.5

In this section, we provide the detailed proof of Theorem 3.5. We first recall the following auxiliary result from (Sonntag & Peitz, 2024).

**Lemma B.1.** *Let $\{h_k\}_{k=1}^K$ be continuously differentiable functions $h_k : \mathbb{R}_{\ge 0} \to \mathbb{R}$, and define $h(t) = \min_{k \in [K]} h_k(t)$. Then:*

*(i) $h$ is differentiable for almost every $t \ge 0$.*

*(ii) for almost every $t \ge 0$, there exists $k \in [K]$ such that $h(t) = h_k(t)$ and $\dot{h}(t) = \dot{h}_k(t)$.*

Let $\{\rho_t\}_{t \ge 0}$ be the solution of the MWGraD flow (9), and let $q \in \mathcal{P}_2(\mathcal{X})$ be an arbitrary reference distribution (e.g., $q \in P_w^*$).

We have the following lemma.

**Lemma B.2.** *$E(\rho_t, q)$ is non-increasing along the MWGraD flow (9), i.e.,*

$$\dot{E}(\rho_t, q) \le 0.$$

*Proof.* For each objective,

$$E_k(\rho_t, q) = F_k(\rho_t) - F_k(q).$$

Differentiating yields

$$\dot{E}_k(\rho_t, q) = \int \langle \nabla \Phi_t, \nabla \delta_\rho F_k(\rho_t) \rangle d\rho_t. \tag{24}$$

By definitions of the MWGraD flow (9) and the projection operator (10),

$$\Phi_t = -h^* = -\arg\min_{h \in \mathcal{C}(\rho_t)} \int \langle \nabla h, \nabla h \rangle d\rho_t = -\arg\min_{h \in \mathcal{C}(\rho_t)} \mathcal{H}(h), \tag{25}$$

where $\mathcal{H}(h) = \int \langle \nabla h, \nabla h \rangle d\rho_t$.

The functional $\mathcal{H}(h)$ is convex with respect to $h$, and its first variation is

$$\delta_h \mathcal{H}(h) = -\nabla \cdot (\rho_t \nabla h).$$

Optimality implies

$$\int \delta_h \mathcal{H}(h^*)(h - h^*) d\mathbf{x} \geq 0,$$

which is equivalent to

$$-\int \nabla \cdot (\rho_t \nabla h^*)(h - h^*) d\mathbf{x} = \int \langle \nabla h - \nabla h^*, \nabla h^* \rangle d\rho_t \geq 0.$$

Choosing $h = \delta_\rho F_k(\rho_t) \in \mathcal{C}(p_t)$ and using $h^* = -\Phi_t$ in (25) gives

$$\int \langle \nabla \Phi_t, \nabla \delta_\rho F_k(\rho_t) \rangle d\rho_t \leq -\int \|\nabla \Phi_t\|_2^2 d\rho_t. \tag{26}$$

Combining (24) and (26) and using Lemma B.1 yields

$$\dot{E}(\rho_t, q) \leq -\int \|\nabla \Phi_t\|_2^2 d\rho_t \leq 0,$$

which concludes the proof. $\qquad\square$

Now we are ready to provide the proof of Theorem 3.5.

*Proof.* Recall the definition of Lyapunov functional

$$\mathcal{V}(q, t) = \frac{1}{2}\mathcal{W}_2^2(\rho_t, q) = \frac{1}{2}\int \|T_t(\mathbf{x}) - \mathbf{x}\|_2^2 d\rho_t(\mathbf{x}),$$

where $T_t$ is the optimal transport mapping from $\rho_t$ to $q$.

Differentiating $\mathcal{V}(q, t)$ with respect to $t$ gives

$$\dot{\mathcal{V}}(q, t) = \int \langle T_t(\mathbf{x}) - \mathbf{x}, \dot{T}_t(\mathbf{x}) \rangle d\rho_t(\mathbf{x}) + \int \langle \nabla \Phi_t(\mathbf{x}), [\nabla T_t - \mathbf{I}_d](T_t(\mathbf{x}) - \mathbf{x}) \rangle d\rho_t(\mathbf{x}), \tag{27}$$

where $\mathbf{I}_d \in \mathbb{R}^{d \times d}$ denotes the identity matrix.

Using the identity from (Wang & Li, 2022), we obtain

$$\int \langle T_t(\mathbf{x}) - \mathbf{x}, \dot{T}_t(\mathbf{x}) \rangle d\rho_t(\mathbf{x}) + \int \langle \nabla \Phi_t(\mathbf{x}), \nabla T_t(\mathbf{x})(T_t(\mathbf{x}) - \mathbf{x}) \rangle d\rho_t(\mathbf{x}) = 0. \tag{28}$$

Using (27) and (28) gives

$$\dot{\mathcal{V}}(q, t) = \int \langle \nabla \Phi_t(\mathbf{x}), \mathbf{x} - T_t(\mathbf{x}) \rangle d\rho_t(\mathbf{x}).$$

By geodesic convexity of each $F_k$ ($k \in [K]$), it follows that

$$F_k(\rho_t) - F_k(q) \leq \int \langle \nabla \delta_\rho F_k(\rho_t)(\mathbf{x}), \mathbf{x} - T_t(\mathbf{x}) \rangle d\rho_t(\mathbf{x}). \tag{29}$$

Multiplying each inequality (29) by $w_{t,k}$ and summing over $k$ yields

$$\sum_{k=1}^{K} w_{t,k}(F_k(\rho_t) - F_k(q)) \leq -\int \langle \nabla \Phi_t(\mathbf{x}), \mathbf{x} - T_t(\mathbf{x}) \rangle d\rho_t(\mathbf{x}) = -\dot{\mathcal{V}}(q, t).$$

Hence

$$E(\rho_t, q) = \min_{k \in [K]} \{F_k(\rho_t) - F_k(q)\} \leq -\dot{\mathcal{V}}(q, t). \tag{30}$$

Integrating (30) over $[0, t]$ gives

$$\int_0^t E(\rho_s, q) ds \leq -(\mathcal{V}(q, t) - \mathcal{V}(q, 0)) \leq \mathcal{V}(q, 0),$$

where we used the fact that $\mathcal{V}(q, t) \geq 0$.

By Lemma B.2, $E(\rho_t, q)$ is non-increasing, so we have

$$E(\rho_t, q) \leq \frac{\mathcal{V}(q, 0)}{t} = \frac{1}{2t}\mathcal{W}_2^2(\rho_0, q).$$

By Assumption 3.2, we further obtain

$$\sup_{\mathbf{F}^* \in \mathbf{F}(P_w^* \cap \Omega_\mathbf{F}(\mathbf{F}(p_0)))} \inf_{q \in \mathbf{F}^{-1}(\{\mathbf{F}^*\})} E(\rho_t, q) \leq \frac{R}{2t}.$$

Finally, from Lemma B.2, $E_k(\rho_t, q) = F_k(\rho_t) - F_k(q)$ is non-increasing, which means that $F_k(\rho_t) \leq F_k(\rho_0)$ for $k \in [K]$. Thus $\rho_t \in \Omega_\mathbf{F}(\mathbf{F}(\rho_0))$. Applying Lemma 3.3 gives

$$\mathcal{M}(\rho_t) = \sup_{q \in \mathcal{P}_2(\mathcal{X})} E(\rho_t, q) \leq \frac{R}{2t}.$$

$\square$

## C. Detailed Proof of Theorem 3.7

In this section, we present the detailed proof of Theorem 3.7.

Let $\{\rho_t\}_{t \geq 0}$ be a solution of the A-MWGraD flow (14). For each $k \in [K]$, define the following global energy

$$W_k(t) = F_k(\rho_t) + \frac{1}{2}\int \|\nabla\Phi_t\|_2^2 d\rho_t.$$

We have the following lemma.

**Lemma C.1.** *For all $k \in [K]$, it follows that*

$$\dot{W}_k(t) \leq -\alpha_t \int \|\nabla\Phi_t\|_2^2 d\rho_t \leq 0.$$

*Proof.* Differentiating $W_k(t)$ with respect to $t$ gives

$$\dot{W}_k(t) = \dot{F}_k(\rho_t) + \frac{1}{2}\int \partial_t\|\nabla\Phi_t\|_2^2 d\rho_t + \frac{1}{2}\int \|\nabla\Phi_t\|_2^2 \dot{\rho}_t d\mathbf{x}$$

$$= \int \langle\nabla\Phi_t, \nabla\delta_\rho F_k(\rho_t)\rangle d\rho_t + \int \langle\nabla\Phi_t, \partial_t\nabla\Phi_t\rangle d\rho_t + \int \langle\nabla\Phi_t, \frac{1}{2}\nabla\|\nabla\Phi_t\|_2^2\rangle d\rho_t$$

$$= \int \langle\nabla\Phi_t, \nabla\delta_\rho F_k(\rho_t) + \partial_t\nabla\Phi_t + \frac{1}{2}\nabla\|\nabla\Phi_t\|_2^2\rangle d\rho_t,$$

where we used the following identity $\dot{\rho}_t = -\nabla \cdot (\rho_t \nabla\Phi_t)$ and applied integration by parts.

Definition of A-MWGraD flow (14) gives

$$-\alpha_t\Phi_t = \dot{\Phi}_t + \frac{1}{2}\|\nabla\Phi_t\|_2^2 + \text{proj}_{\mathcal{C}(\rho_t), \rho_t}[0] = \text{proj}_{\mathcal{C}(\rho_t) + \dot{\Phi}_t + \frac{1}{2}\|\nabla\Phi_t\|_2^2, \rho_t}[0]$$

$$= \underset{h \in \mathcal{C}(\rho_t) + \dot{\Phi}_t + \frac{1}{2}\|\nabla\Phi_t\|_2^2}{\arg\min} \mathcal{G}(h),$$

where $\mathcal{G}(h) = \int \langle\nabla h, \nabla h\rangle d\rho_t$ is the convex function with respect to $h$. Optimality gives

$$\int \delta_h G(h^*)(h - h^*) d\mathbf{x} = \int \langle\nabla h^*, \nabla h - \nabla h^*\rangle d\rho_t \geq 0.$$

Choosing $h = \delta_\rho F_k(\rho_t) + \dot{\Phi}_t + \frac{1}{2}\|\nabla\Phi_t\|_2^2$ and using $h^* = -\alpha_t\Phi_t$, we have

$$\int \langle\nabla\Phi_t, \nabla\delta_\rho F_k(\rho_t) + \partial_t\nabla\Phi_t + \frac{1}{2}\nabla\|\nabla\Phi_t\|_2^2\rangle d\rho_t \leq -\alpha_t \int \|\nabla\Phi_t\|_2^2 d\rho_t, \tag{31}$$

which concludes the proof. $\square$

Due to the effect of damping term $-\alpha_t \Phi_t$, it is not guaranteed that $F_k(\rho_t)$ is non-increasing along the A-MWGraD flow (14). However, the following corollary guarantees that $F_k(\rho_t)$ is upper bounded by the initial functional value $F_k(\rho_0)$ given $\nabla \Phi_0 = 0$.

**Corollary C.2.** *Let $\{\rho_t\}_{t \geq 0}$ be a solution of the A-MWGraD flow (14) with $\nabla \Phi_0 = 0$. For $k \in [K]$, it holds that $F_k(\rho_t) \leq F_k(\rho_0)$, i.e., $\rho_t \in \Omega_{\boldsymbol{F}}(\boldsymbol{F}(\rho_0))$, for all $t \geq 0$.*

*Proof.* From Lemma C.1, it follows that

$$F_k(\rho_0) = W_k(\rho_0) \geq F_k(\rho_t) + \frac{1}{2} \int \|\nabla \Phi_t\|_2^2 d\rho_t \geq F_k(\rho_t).$$

$\square$

Now we are ready to present the proof of Theorem 3.7. We consider two cases for objectives $F_k$ ($k \in [K]$): geodesically convex and $\beta$-strongly geodesically convex.

Consider $F_k$ is geodesically convex for all $k \in [K]$ and define

$$\mathcal{E}_{k,\lambda}(\rho_t, q) = t^2(F_k(\rho_t) - F_k(q)) + \frac{1}{2} \int \|2(\mathbf{x} - T_t(\mathbf{x})) + t\nabla \Phi_t(\mathbf{x})\|_2^2 d\rho_t(\mathbf{x})$$
$$+ \frac{\lambda}{2} \int \|\mathbf{x} - T_t(\mathbf{x})\|_2^2 d\rho_t(\mathbf{x}),$$

and

$$\mathcal{E}_\lambda(\rho_t, q) = \min_{k \in [K]} \mathcal{E}_{k,\lambda}(\rho_t, q).$$

We investigate the non-increasing of $\mathcal{E}_\lambda(\rho_t, q)$ through the following lemma.

**Lemma C.3.** *Suppose $F_k$ is geodesically convex for $k \in [K]$. Let $\lambda = 2(\alpha - 3)$ and $\alpha \geq 3$. Then, it follows that*

$$\dot{\mathcal{E}}_{k,\lambda}(\rho_t, q) \leq 2t(F_k(\rho_t) - F_k(q)) - 2t \min_{l \in [K]} (F_l(\rho_t) - F_l(q)) + t(3 - \alpha) \int \|\nabla \Phi_t\|_2^2 d\rho_t. \tag{32}$$

*Proof.* Differentiating $\mathcal{E}_{k,\lambda}(t)$ with respect to $t$ gives

$$\dot{\mathcal{E}}_{k,\lambda}(\rho_t, q) = 2t(F_k(\rho_t) - F_k(q)) - t^2 \int \delta_\rho F_k(\rho_t) \nabla \cdot (\rho_t \nabla \Phi_t) d\mathbf{x}$$
$$+ 4 \int \langle \partial_t T_t(\mathbf{x}), T_t(\mathbf{x}) - \mathbf{x} \rangle d\rho_t(\mathbf{x}) - 2 \int \|\mathbf{x} - T_t(\mathbf{x})\|_2^2 \nabla \cdot (\rho_t \nabla \Phi_t) d\mathbf{x}$$
$$+ t \int \langle \nabla \Phi_t, \nabla \Phi_t + t\partial_t \nabla \Phi_t \rangle d\rho_t - \frac{1}{2} t^2 \int \|\nabla \Phi_t\|_2^2 \nabla \cdot (\rho_t \nabla \Phi_t) d\mathbf{x}$$
$$- 2 \int \langle \partial_t T_t, t\nabla \Phi_t \rangle d\rho_t + 2 \int \langle \mathbf{x} - T_t(\mathbf{x}), \nabla \Phi_t(\mathbf{x}) + t\partial_t \nabla \Phi_t(\mathbf{x}) \rangle d\rho_t(\mathbf{x})$$
$$- 2t \int \langle \mathbf{x} - T_t(\mathbf{x}), \nabla \Phi_t(\mathbf{x}) \rangle \nabla \cdot (p_t \nabla \Phi_t) d\mathbf{x}$$
$$+ \lambda \int \langle \partial_t T_t(\mathbf{x}), T_t(\mathbf{x}) - \mathbf{x} \rangle d\rho_t(\mathbf{x}) - \lambda \int \|\mathbf{x} - T_t(\mathbf{x})\|_2^2 \nabla \cdot (\rho_t \nabla \Phi_t) d\mathbf{x}.$$

Here, we used the identity $\dot{\rho}_t = -\nabla \cdot (\rho_t \nabla \Phi_t)$.

By integration by parts, we obtain

$$
\begin{aligned}
\dot{\mathcal{E}}_{k,\lambda}(\rho_t, q) = {}& 2t(F_k(\rho_t) - F_k(q)) + t^2 \int \langle \nabla\Phi_t, \nabla\delta_\rho F_k(\rho_t)\rangle d\rho_t \\
& + 4\int \langle \partial_t T_t(\mathbf{x}), T_t(\mathbf{x}) - \mathbf{x}\rangle d\rho_t(\mathbf{x}) + 4\int \langle \nabla\Phi_t(\mathbf{x}), [\mathbf{I}_d - \nabla T_t(\mathbf{x})](\mathbf{x} - T_t(\mathbf{x}))\rangle d\rho_t(\mathbf{x}) \\
& + t\int \|\nabla\Phi_t\|_2^2 d\rho_t + t^2\int \langle \nabla\Phi_t, \partial_t\nabla\Phi_t\rangle d\rho_t + t^2\int \langle \nabla\Phi_t, \tfrac{1}{2}\nabla\|\nabla\Phi_t\|_2^2\rangle d\rho_t \\
& - 2t\int \langle \nabla\Phi_t, \partial_t T_t\rangle d\rho_t + 2\int \langle \mathbf{x} - T_t(\mathbf{x}), \nabla\Phi_t(\mathbf{x}) + t\partial_t\nabla\Phi_t(\mathbf{x})\rangle d\rho_t(\mathbf{x}) \\
& + 2t\int \langle \nabla\Phi_t(\mathbf{x}), [\mathbf{I}_d - \nabla T_t(\mathbf{x})]\nabla\Phi_t(\mathbf{x})\rangle d\rho_t(\mathbf{x}) + 2t\int \langle \nabla\Phi_t(\mathbf{x}), [\nabla^2\Phi_t(\mathbf{x})](\mathbf{x} - T_t(\mathbf{x}))\rangle d\rho_t(\mathbf{x}) \\
& + \lambda\int \langle \partial_t T_t(\mathbf{x}), T_t(\mathbf{x}) - \mathbf{x}\rangle\rho_t(\mathbf{x}) + \lambda\int \langle \nabla\Phi_t(\mathbf{x}), [\mathbf{I}_d - \nabla T_t(\mathbf{x})](\mathbf{x} - T_t(\mathbf{x}))\rangle d\rho_t(\mathbf{x}).
\end{aligned}
$$

Rearranging the terms above gives

$$
\begin{aligned}
\dot{\mathcal{E}}_{k,\lambda}(\rho_t, q) = {}& 2t(F_k(\rho_t) - F_k(q)) + t^2\int \langle \nabla\Phi_t, \nabla\delta_\rho F_k(\rho_t) + \tfrac{1}{2}\nabla\|\nabla\Phi_t\|_2^2 + \partial_t\nabla\Phi_t\rangle d\rho_t \\
& + t\int \langle T_t(\mathbf{x}) - \mathbf{x}, -\frac{6+\lambda}{t}\nabla\Phi_t - 2\partial_t\nabla\Phi_t - \nabla\|\nabla\Phi_t\|_2^2\rangle d\rho_t(\mathbf{x}) \\
& + 3t\int \|\nabla\Phi_t\|_2^2 d\rho_t - 2t\int \langle \nabla\Phi_t, \nabla T_t\nabla\Phi_t\rangle d\rho_t - 2t\int \langle \nabla\Phi_t, \partial T_t\rangle d\rho_t \\
& + (4+\lambda)\left(\int \langle \nabla\Phi_t(\mathbf{x}), [\nabla T_t(\mathbf{x})](T_t(\mathbf{x}) - \mathbf{x})\rangle d\rho_t(\mathbf{x}) + \int \langle \partial T_t(\mathbf{x}), T_t(\mathbf{x}) - \mathbf{x}\rangle d\rho_t(\mathbf{x})\right).
\end{aligned}
\tag{33}
$$

Next, we have the following useful equality and inequality in (Wang & Li, 2022):

$$
\int \langle \partial_t T_t, \nabla\Phi_t\rangle d\rho_t(\mathbf{x}) + \int \langle \nabla\Phi_t, \nabla T_t\nabla\Phi_t\rangle d\rho_t(\mathbf{x}) \geq 0,
\tag{34}
$$

$$
\int \langle \partial_t T_t(\mathbf{x}), T_t(\mathbf{x}) - \mathbf{x}\rangle d\rho_t(\mathbf{x}) + \int \langle \nabla\Phi_t(\mathbf{x}), \nabla T_t(\mathbf{x})(T_t(\mathbf{x}) - \mathbf{x})\rangle d\rho_t(\mathbf{x}) = 0.
\tag{35}
$$

By (34) and (35), we can simplify (33) as follows

$$
\begin{aligned}
\dot{\mathcal{E}}_{k,\lambda}(\rho_t, q) = {}& 2t(F_k(\rho_t) - F_k(q)) + t^2\int \langle \nabla\Phi_t, \nabla\delta_\rho F_k(\rho_t) + \nabla\|\nabla\Phi_t\|_2^2 + \partial_t\nabla\Phi_t\rangle d\rho_t(\mathbf{x}) \\
& + t\int \langle T_t(\mathbf{x}) - \mathbf{x}, -\frac{6+\lambda}{t}\nabla\Phi_t - 2\partial_t\nabla\Phi_t - \nabla\|\nabla\Phi_t\|_2^2\rangle d\rho_t(\mathbf{x}) \\
& + 3t\int \|\nabla\Phi_t\|_2^2 d\rho_t(\mathbf{x}).
\end{aligned}
\tag{36}
$$

By definition of the A-MWGraD flow (14), we have

$$
-\partial_t\nabla\Phi_t - \frac{\alpha}{t}\nabla\Phi_t - \frac{1}{2}\nabla\|\nabla\Phi_t\|_2^2 = \sum_{k=1}^{K} w_{t,k}\nabla\delta_\rho F_k(\rho_t).
\tag{37}
$$

By using (31) of Lemma C.1, $\lambda = 2(\alpha - 3)$, and (37), we can upper bound (36) as follows

$$
\dot{\mathcal{E}}_{k,\lambda}(\rho_t, q) \leq 2t(F_k(\rho_t) - F_k(q)) - \alpha t \int \|\nabla \Phi_t\|_2^2 d\rho_t
$$
$$
+ 2t \int \langle T_t(\mathbf{x}) - \mathbf{x}, \sum_{k=1}^{K} w_{t,k} \nabla \delta_\rho F_k(\rho_t)(\mathbf{x}) \rangle d\rho_t(\mathbf{x})
$$
$$
+ 3t \int \|\nabla \Phi_t\|_2^2 d\rho_t
$$
$$
\leq 2t(F_k(\rho_t) - F_k(q)) - 2t \min_{l \in [K]} (F_l(\rho_t) - F_l(q)) + t(3 - \alpha) \int \|\nabla \Phi_t\|_2^2 d\rho_t,
$$

which conclude the proof of Lemma C.3. $\qquad \square$

Consider $F_k$ is $\beta$-strongly geodesically convex for $k \in [K]$. Motivated by (Taghvaei & Mehta, 2019; Wang & Li, 2022), we define the following function

$$
\mathcal{E}_k(\rho_t, q) = e^{\sqrt{\beta}t}(F_k(\rho_t) - F_k(q)) + \frac{e^{\sqrt{\beta}t}}{2} \int \|\sqrt{\beta}(\mathbf{x} - T_t(\mathbf{x})) + \nabla \Phi_t(\mathbf{x})\|_2^2 d\rho_t(\mathbf{x}),
$$

and

$$
\mathcal{E}(\rho_t, q) = \min_{k \in [K]} \mathcal{E}_k(\rho_t, q).
$$

We investigate the non-increasing of $\mathcal{E}(\rho_t, q)$ through the following lemma.

**Lemma C.4.** *Suppose $F_k$ is $\beta$-strongly geodesically convex for $k \in [K]$. Let $\alpha_t = 2\sqrt{\beta}$. Then it follows that*

$$
e^{-\sqrt{\beta}t}\dot{\mathcal{E}}_k(\rho_t, q) \leq \sqrt{\beta}\mathcal{G}_k(\rho_t, q) - \sqrt{\beta} \min_{l \in [K]} \mathcal{G}_l(\rho_t, q) - \frac{\sqrt{\beta}}{2} \int \|\nabla \Phi_t\|_2^2 d\rho_t, \tag{38}
$$

*where $\mathcal{G}_k(\rho_t, q) = - \int \langle T_t(\mathbf{x}) - \mathbf{x}, \nabla \delta_\rho F_k(\rho_t)(\mathbf{x}) \rangle d\rho_t(\mathbf{x})$.*

*Proof.* Differentiating $\mathcal{E}_k(\rho_t, q)$ with respect to $t$ gives

$$
e^{-\sqrt{\beta}t}\dot{\mathcal{E}}_k(\rho_t, q) = \sqrt{\beta}(F_k(\rho_t) - F_k(q)) - \int \delta_\rho F_k(\rho_t) \nabla \cdot (\rho_t \nabla \Phi_t) d\mathbf{x}
$$
$$
+ \frac{\sqrt{\beta^3}}{2} \int \|\mathbf{x} - T_t(\mathbf{x})\|_2^2 d\rho_t(\mathbf{x}) + \frac{\sqrt{\beta}}{2} \int \|\nabla \Phi_t\|_2^2 d\rho_t + \beta \int \langle \mathbf{x} - T_t(\mathbf{x}), \nabla \Phi_t(\mathbf{x}) \rangle d\rho_t(\mathbf{x})
$$
$$
+ \beta \int \langle T_t(\mathbf{x}) - \mathbf{x}, \partial_t T_t(\mathbf{x}) \rangle d\rho_t(\mathbf{x}) - \frac{\beta}{2} \int \|\mathbf{x} - T_t(\mathbf{x})\|_2^2 \nabla \cdot (\rho_t \nabla \Phi_t) d\mathbf{x}
$$
$$
+ \int \langle \nabla \Phi_t(\mathbf{x}), \partial_t \nabla \Phi_t \rangle d\rho_t(\mathbf{x}) - \frac{1}{2} \int \|\nabla \Phi_t(\mathbf{x})\|_2^2 \nabla \cdot (\rho_t \nabla \Phi_t) d\mathbf{x} \tag{39}
$$
$$
- \sqrt{\beta} \int \langle \partial_t T_t(\mathbf{x}), \nabla \Phi_t(\mathbf{x}) \rangle d\rho_t(\mathbf{x}) - \sqrt{\beta} \int \langle T_t(\mathbf{x}) - \mathbf{x}, \partial_t \nabla \Phi_t(\mathbf{x}) \rangle d\rho_t(\mathbf{x})
$$
$$
+ \sqrt{\beta} \int \langle T_t(\mathbf{x}) - \mathbf{x}, \nabla \Phi_t(\mathbf{x}) \rangle \nabla \cdot (\rho_t \nabla \Phi_t) d\mathbf{x}
$$

By $\beta$-strongly geodesic convexity of $F_k$ for $k \in [K]$, we obtain

$$
\frac{\sqrt{\beta^3}}{2} \int \|\mathbf{x} - T_t(\mathbf{x})\|_2^2 d\rho_t(\mathbf{x}) + \sqrt{\beta}(F_k(\rho_t) - F_k(q)) \leq -\sqrt{\beta} \int \langle T_t(\mathbf{x}) - \mathbf{x}, \nabla \delta_\rho F_k(\rho_t)(\mathbf{x}) \rangle d\rho_t(\mathbf{x}) \tag{40}
$$

Using the identity $\dot{\rho}_t = -\nabla \cdot (\rho_t \nabla \Phi_t)$ gives

$$-\int \delta_\rho F_k(\rho_t) \nabla \cdot (\rho_t \nabla \Phi_t) d\mathbf{x} + \int \langle \nabla \Phi_t(\mathbf{x}), \partial_t \nabla \Phi_t(\mathbf{x}) \rangle d\rho_t(\mathbf{x})$$

$$-\frac{1}{2} \int \|\nabla \Phi_t\|_2^2 \nabla \cdot (\rho_t \nabla \Phi_t) d\mathbf{x}$$

$$= \int \langle \Phi_t(\mathbf{x}), \nabla \delta_\rho F_k(\rho_t)(\mathbf{x}) + \partial_t \nabla \Phi_t(\mathbf{x}) + \frac{1}{2} \nabla \|\nabla \Phi_t(\mathbf{x})\|_2^2 \rangle d\rho_t(\mathbf{x})$$

$$\leq -\alpha_t \int \|\nabla \Phi_t(\mathbf{x})\|_2^2 d\rho_t(\mathbf{x}) = -2\sqrt{\beta} \int \|\nabla \Phi_t(\mathbf{x})\|_2^2 d\rho_t(\mathbf{x}),$$

where we used (31) of the proof of Lemma C.1. We also have

$$-\frac{\beta}{2} \int \|\mathbf{x} - T_t(\mathbf{x})\|_2^2 \nabla \cdot (\rho_t \nabla \Phi_t) d\mathbf{x} = \beta \int \langle \nabla \Phi_t(\mathbf{x}), [\nabla T_t(\mathbf{x}) - \mathbf{I}_d] (T_t(\mathbf{x}) - \mathbf{x}) \rangle d\rho_t(\mathbf{x})$$

$$= \beta \int \langle \nabla \Phi_t(\mathbf{x}), [\nabla T_t(\mathbf{x})] (T_t(\mathbf{x}) - \mathbf{x}) \rangle d\rho_t(\mathbf{x}) - \beta \int \langle \nabla \Phi_t(\mathbf{x}), T_t(\mathbf{x}) - \mathbf{x} \rangle d\rho_t(\mathbf{x}). \tag{41}$$

$$\sqrt{\beta} \int \langle T_t(\mathbf{x}) - \mathbf{x}, \nabla \Phi_t(\mathbf{x}) \rangle \nabla \cdot (\rho_t \nabla \Phi_t) d\mathbf{x} = -\sqrt{\beta} \int \langle \nabla \Phi_t(\mathbf{x}), [\nabla T_t(\mathbf{x}) - \mathbf{I}] \nabla \Phi_t(\mathbf{x}) \rangle d\rho_t(\mathbf{x})$$

$$-\sqrt{\beta} \int \langle \nabla \Phi_t(\mathbf{x}), \nabla^2 \Phi_t(\mathbf{x})(T_t(\mathbf{x}) - \mathbf{x}) \rangle d\rho_t(\mathbf{x})$$

$$= -\sqrt{\beta} \int \langle T_t(\mathbf{x}) - \mathbf{x}, \frac{1}{2} \nabla \|\nabla \Phi_t(\mathbf{x})\|_2^2 \rangle d\rho_t(\mathbf{x}) - \sqrt{\beta} \int \langle \nabla \Phi_t(\mathbf{x}), \nabla T_t(\mathbf{x}) \nabla \Phi_t(\mathbf{x}) \rangle d\rho_t(\mathbf{x}) \tag{42}$$

$$+ \sqrt{\beta} \int \|\nabla \Phi_t(\mathbf{x})\|_2^2 d\rho_t(\mathbf{x})$$

By (34), (35), (40), (41) and (42), we can simplify (39) as follows

$$e^{-\sqrt{\beta}t} \dot{\mathcal{E}}_k(\rho_t, q) \leq -\sqrt{\beta} \int \langle T_t(\mathbf{x}) - \mathbf{x}, \nabla \delta_\rho F_k(\rho_t) \rangle d\rho_t(\mathbf{x}) - \frac{\sqrt{\beta}}{2} \int \|\nabla \Phi_t(\mathbf{x})\|_2^2 d\rho_t(\mathbf{x})$$

$$+ \underbrace{\beta \int \langle \partial_t T_t(\mathbf{x}), T_t(\mathbf{x}) - \mathbf{x} \rangle d\rho_t(\mathbf{x}) + \beta \int \langle \nabla \Phi_t(\mathbf{x}), [\nabla T_t(\mathbf{x})](T_t(\mathbf{x}) - \mathbf{x}) \rangle d\rho_t(\mathbf{x})}_{=0}$$

$$- \underbrace{\sqrt{\beta} \left( \int \langle \partial_t T_t, \nabla \Phi_t \rangle d\rho_t(\mathbf{x}) + \int \langle \nabla \Phi_t, \nabla T_t \nabla \Phi_t \rangle d\rho_t(\mathbf{x}) \right)}_{\geq 0}$$

$$- \sqrt{\beta} \int \langle T_t(\mathbf{x}) - \mathbf{x}, \frac{1}{2} \nabla \|\nabla \Phi_t(\mathbf{x})\|_2^2 + 2\sqrt{\beta} \nabla \Phi_t(\mathbf{x}) + \partial_t \nabla \Phi_t(\mathbf{x}) \rangle d\rho_t(\mathbf{x})$$

$$= -\sqrt{\beta} \int \langle T_t(\mathbf{x}) - \mathbf{x}, \nabla \delta_\rho F_k(\rho_t) \rangle d\rho_t(\mathbf{x}) + \sqrt{\beta} \int \langle T_t(\mathbf{x}) - \mathbf{x}, \sum_{l=1}^K w_{t,l} \nabla \delta_\rho F_l(\rho_t) \rangle d\rho_t(\mathbf{x}) - \frac{\sqrt{\beta}}{2} \int \|\nabla \Phi_t(\mathbf{x})\|_2^2 d\rho_t(\mathbf{x})$$

$$= \sqrt{\beta} \mathcal{G}_k(\rho_t, q) - \sqrt{\beta} \sum_{l=1}^K w_{t,l} \mathcal{G}_l(\rho_t, q) - \frac{\sqrt{\beta}}{2} \int \|\nabla \Phi_t(\mathbf{x})\|_2^2 d\rho_t(\mathbf{x})$$

$$\leq \sqrt{\beta} \mathcal{G}_k(\rho_t, q) - \sqrt{\beta} \min_{l \in [K]} \mathcal{G}_l(\rho_t, q) - \frac{\sqrt{\beta}}{2} \int \|\nabla \Phi_t(\mathbf{x})\|_2^2 d\rho_t(\mathbf{x})$$

$$\tag{43}$$

which concludes the proof of Lemma C.4.

$\square$

Now we are ready to present the proof of Theorem 3.7 using Lemma C.3 and Lemma C.4.

*Proof.* We first focus on proving (16) for the geodesically convex case. From Lemma B.1, we can see that for $t \geq 0$, then there exists $k$ such that $\mathcal{E}_\lambda(\rho_t, q) = \mathcal{E}_{k,\lambda}(\rho_t, q)$ and $\dot{\mathcal{E}}_\lambda(t) = \dot{\mathcal{E}}_{k,\lambda}(t)$. Thus, by Lemma C.3, we obtain

$$\dot{\mathcal{E}}_\lambda(\rho_t, q) = \dot{\mathcal{E}}_{k,\lambda}(\rho_t, q) \leq 2t(F_k(\rho_t) - F_k(q)) - 2t(F_k(\rho_t) - F_k(q)) + t(3 - \alpha) \int \|\nabla \Phi_t\|_2^2 d\rho_t(\mathbf{x}).$$

Choosing $\alpha \geq 3$ gives: $\dot{\mathcal{E}}_\lambda(\rho_t, q) \leq 0$, which confirms its non-increasing. Thus, we obtain

$$t^2 \min_{k \in [K]} (F_k(\rho_t) - F_k(q)) \leq \mathcal{E}_\lambda(\rho_t, q) \leq \mathcal{E}_\lambda(\rho_0, q) = (\alpha - 1) \int \|\mathbf{x} - T_t(\mathbf{x})\|_2^2 d\rho_t(\mathbf{x})$$
$$= (\alpha - 1)\mathcal{W}_2^2(\rho_0, q).$$

We follow similar steps in proof of Theorem 3.5. By Assumption 3.2, we obtain

$$\sup_{\mathbf{F}^* \in \mathbf{F}(P_w^* \cap \Omega_\mathbf{F}(\mathbf{F}(p_0)))} \inf_{q \in \mathbf{F}^{-1}(\{\mathbf{F}^*\})} \min_{k \in [K]} (F_k(p_t) - F_k(q)) \leq \frac{(\alpha - 1)R}{t^2}.$$

We get

$$\sup_{\mathbf{F}^* \in \mathbf{F}(P_w^* \cap \Omega_\mathbf{F}(\mathbf{F}(\rho_0)))} \min_{k \in [K]} \{F_k(\rho_t) - F_k^*\} \leq \frac{(\alpha - 1)R}{t^2},$$

which gives

$$\sup_{q \in P_w^* \cap \Omega_\mathbf{F}(\mathbf{F}(\rho_0))} \min_{k \in [K]} \{F_k(\rho_t) - F_k(q)\} \leq \frac{(\alpha - 1)R}{t^2}.$$

Furthermore, as shown in Corollary C.2, $F_k(\rho_t) \leq F_k(\rho_0)$ for all $k$, which ensures that $\rho_t \in \Omega_\mathbf{F}(\mathbf{F}(\rho_0))$. Applying Lemma 3.3, we conclude the proof of (16).

Next we focus on proving (17) for the $\beta$-strongly geodesically convex case. Similarly, from Lemma B.1, we have that for $t \geq 0$, there exists $k$ such that $\mathcal{E}(\rho_t, q) = \mathcal{E}_k(\rho_t, q)$ and $\dot{\mathcal{E}}(\rho_t, q) = \dot{\mathcal{E}}_k(\rho_t, q)$. Thus, by Lemma C.4, we obtain

$$e^{-\sqrt{\beta}t} \dot{\mathcal{E}}(\rho_t, q) = e^{-\sqrt{\beta}t} \min_{l \in [K]} \dot{\mathcal{E}}_l(\rho_t, q) \leq -\frac{\sqrt{\beta}}{2} \int \|\nabla \Phi_t(\mathbf{x})\|_2^2 d\rho_t(\mathbf{x}) \leq 0.$$

This gives: $\dot{\mathcal{E}}(\rho_t, q) \leq 0$, which confirms its non-increasing. Thus, we obtain

$$\min_{k \in [K]} \{F_k(\rho_t) - F_k(q)\} \leq e^{-\sqrt{\beta}t} \left[ \min_{l \in [K]} \{F_l(\rho_0) - F_l(q)\} + \frac{\beta}{2} \mathcal{W}_2^2(\rho_0, q) \right].$$

Here we remark that for any $q \in P_w^* \cap \Omega_\mathbf{F}(\mathbf{F}(\rho_0))$, we get $\min_{l \in [K]} \{F_l(\rho_0) - F_l(q)\} \leq \min_{l \in [K]} F_l(\rho_0)$ and $\mathcal{W}_2^2(\rho_0, q) \leq R$. Hence,

$$\sup_{\mathbf{F}^* \in \mathbf{F}(P_w^* \cap \Omega_\mathbf{F}(\mathbf{F}(p_0)))} \inf_{q \in \mathbf{F}^{-1}(\{\mathbf{F}^*\})} \min_{k \in [K]} (F_k(\rho_t) - F_k(q)) \leq e^{-\sqrt{\beta}t} \left[ \min_{l \in [K]} F_l(\rho_0) + \frac{\beta R}{2} \right].$$

Finally, applying Lemma 3.3, we concludes the proof of (17).

$\square$

# D. Detailed Proofs of Theorem 3.1 and Theorem 3.6

We first collect several identities that will be used throughout the proofs.

Suppose $\rho = \mathcal{N}(0, \Sigma)$ and $\pi_k = \mathcal{N}(0, \Sigma_*^k)$ for $k \in [K]$,

$$\delta_\rho F_k(\rho)(\mathbf{x}) = \frac{1}{2} \log \frac{\det \Sigma_*^k}{\det \Sigma} + \frac{1}{2} \mathbf{x}^\top \left[ (\Sigma_*^k)^{-1} - \Sigma^{-1} \right] \mathbf{x} + 1, \tag{44}$$

$$\nabla \delta_\rho F_k(\rho)(\mathbf{x}) = \left[ \left( \Sigma_*^k \right)^{-1} - \Sigma^{-1} \right] \mathbf{x}, \tag{45}$$

$$\nabla_\Sigma F_k^\dagger(\Sigma) = \frac{1}{2} \left[ \left( \Sigma_*^k \right)^{-1} - \Sigma^{-1} \right]. \tag{46}$$

We next state two auxiliary lemmas, that will be used to establish the positive-definiteness of the covariance matrix, i.e. $\Sigma_t \succ 0$ for all $t \geq 0$.

**Lemma D.1.** *Suppose* $\mathbf{A}, \mathbf{B} \in \mathbb{R}^{d \times d}$ *are positive definite matrices, i.e.,* $\mathbf{A} \succ 0, \mathbf{B} \succ 0$. *Then* $\mathbf{A} \left( \mathbf{B} \right)^{-1}$ *shares eigenvalues with* $\mathbf{A}^{\frac{1}{2}} \left( \mathbf{B} \right)^{-1} \mathbf{A}^{\frac{1}{2}}$.

*Proof.* Suppose $\lambda$ is an eigenvalue of $\mathbf{A}^{\frac{1}{2}} \left( \mathbf{B} \right)^{-1} \mathbf{A}^{\frac{1}{2}}$ with eigenvector $\mathbf{v} \neq 0$, i.e.,

$$\mathbf{A}^{\frac{1}{2}} \left( \mathbf{B} \right)^{-1} \mathbf{A}^{\frac{1}{2}} \mathbf{v} = \lambda \mathbf{v}.$$

Multiplying on the left by $\mathbf{A}^{\frac{1}{2}}$ yields

$$\mathbf{A} \left( \mathbf{B} \right)^{-1} \mathbf{A}^{\frac{1}{2}} \mathbf{v} = \lambda \left( \mathbf{A}^{\frac{1}{2}} \mathbf{v} \right).$$

As $\mathbf{A} \succ 0$, its square root is invertible, so $\mathbf{A}^{\frac{1}{2}} \mathbf{v} \neq 0$. Hence $\lambda$ is an eigenvalue of $\mathbf{A} \left( \mathbf{B} \right)^{-1}$. $\square$

**Lemma D.2** (Eigenvalue Continuity). *Let* $\mathbf{A}_t \in \mathbb{R}^{d \times d}$ *be a family of symmetric matrices whose entries* $[\mathbf{A}_t]_{ij}$ *are continuous functions of* $t$. *Then* $t \mapsto \lambda_{\min}(\mathbf{A}_t)$ *is continuous, where* $\lambda_{\min}(\mathbf{A})$ *denotes the smallest eigenvalue of* $\mathbf{A}$.

*Proof.* Fix any $t_0$. We show $|\lambda_{\min}(\mathbf{A}_t) - \lambda_{\min}(\mathbf{A}_{t_0})| \to 0$ as $t \to t_0$. Since $\mathbf{A}_t$ is symmetric,

$$\lambda_{\min}(\mathbf{A}_t) = \min_{\|\mathbf{v}\|_2 = 1} \mathbf{v}^\top \mathbf{A}_t \mathbf{v}.$$

Let $\mathbf{v}_t^* = \arg\min_{\|\mathbf{v}\|_2 = 1} \mathbf{v}^\top \mathbf{A}_t \mathbf{v}$ and $\mathbf{v}_0^* = \arg\min_{\|\mathbf{v}\|_2 = 1} \mathbf{v}^\top \mathbf{A}_{t_0} \mathbf{v}$. By optimality of each minimizer,

$$\mathbf{v}_t^{*\top} \mathbf{A}_t \mathbf{v}_t^* \leq \mathbf{v}_0^{*\top} \mathbf{A}_t \mathbf{v}_0^*, \quad \mathbf{v}_0^{*\top} \mathbf{A}_{t_0} \mathbf{v}_0^* \leq \mathbf{v}_t^{*\top} \mathbf{A}_{t_0} \mathbf{v}_t^*. \tag{47}$$

Subtracting these inequalities yields

$$\lambda_{\min}(\mathbf{A}_t) - \lambda_{\min}(\mathbf{A}_{t_0}) \leq \mathbf{v}_0^{*\top} \left( \mathbf{A}_t - \mathbf{A}_{t_0} \right) \mathbf{v}_0^*,$$
$$\lambda_{\min}(\mathbf{A}_{t_0}) - \lambda_{\min}(\mathbf{A}_t) \leq \mathbf{v}_t^{*\top} \left( \mathbf{A}_{t_0} - \mathbf{A}_t \right) \mathbf{v}_t^*.$$

For any unit vector $\mathbf{v}$, by Cauchy–Schwarz inequality, we obtain

$$|\mathbf{v}^\top \left( \mathbf{A}_t - \mathbf{A}_{t_0} \right) \mathbf{v}| \leq \|(\mathbf{A}_t - \mathbf{A}_{t_0})\mathbf{v}\|_2 \|\mathbf{v}\|_2 \leq \|\mathbf{A}_t - \mathbf{A}_{t_0}\|_{\mathrm{op}} \|\mathbf{v}\|_2^2 = \|\mathbf{A}_t - \mathbf{A}_{t_0}\|_{\mathrm{op}},$$

where $\|\mathbf{A}\|_{\mathrm{op}}$ is the operator norm of matrix $\mathbf{A}$. Therefore,

$$|\lambda_{\min}(\mathbf{A}_t) - \lambda_{\min}(\mathbf{A}_{t_0})| \leq \|\mathbf{A}_t - \mathbf{A}_{t_0}\|_{\mathrm{op}}.$$

Since $\|\mathbf{A}_t - \mathbf{A}_{t_0}\|_{\mathrm{op}} \to 0$ as $t \to 0$, it follows that $|\lambda_{\min}(\mathbf{A}_t) - \lambda_{\min}(\mathbf{A}_{t_0})| \to 0$. This establishes the continuity of $\lambda_{\min}(\mathbf{A}_t)$ $\square$

Now we present the proofs of Theorem 3.1 and Theorem 3.6.

*Proof.* (of Theorem 3.1) Assuming $\Sigma_t \succ 0$, we establish the equivalence between the MWGraD flow (9) and the covariance dynamics (11). We record two standard matrix calculus identities that will be used repeatedly below (Wang & Li, 2022).

$$\frac{\partial}{\partial t} \det(\Sigma_t) = \det(\Sigma_t) \operatorname{tr}(\Sigma_t^{-1} \dot{\Sigma}_t), \tag{48}$$

$$\frac{\partial}{\partial t} \Sigma_t^{-1} = -\Sigma_t^{-1} \dot{\Sigma}_t \Sigma_t^{-1}. \tag{49}$$

Combining with $\dot{\Sigma}_t = 2(S_t \Sigma_t + \Sigma_t S_t)$ yields

$$\operatorname{tr}(\Sigma_t^{-1} \dot{\Sigma}_t) = 2 \operatorname{tr}(S_t + \Sigma_t^{-1} S_t \Sigma_t) = 4 \operatorname{tr}(S_t), \tag{50}$$

$$\operatorname{tr}(\mathbf{x}^\top \Sigma_t^{-1} \dot{\Sigma}_t \Sigma_t^{-1} \mathbf{x}) = 2 \operatorname{tr}(\mathbf{x}^\top \Sigma_t^{-1} S_t \mathbf{x} + \mathbf{x}^\top S_t \Sigma_t^{-1} \mathbf{x}) = 4 \operatorname{tr}(S_t \Sigma_t^{-1} \mathbf{x} \mathbf{x}^\top). \tag{51}$$

**Continuity equation**: $\dot{\rho}_t + \nabla \cdot (\rho_t \nabla \Phi_t) = 0$.

Differentiating $\rho_t$ using (48)-(51) yields

$$\begin{aligned}
\dot{\rho}_t &= \frac{\partial}{\partial t}\left(\frac{1}{\sqrt{\det \Sigma_t}}\right)\sqrt{\det \Sigma_t}\rho_t + \frac{1}{2}\operatorname{tr}(\mathbf{x}\Sigma_t^{-1}\dot{\Sigma}_t\Sigma_t^{-1}\mathbf{x})\rho_t \\
&= -\frac{1}{2}\operatorname{tr}(\Sigma_t^{-1}\dot{\Sigma}_t)\rho_t + 2\operatorname{tr}(S_t\Sigma_t^{-1}\mathbf{x}\mathbf{x}^\top)\rho_t \\
&= -2\operatorname{tr}(S_t)\rho_t + 2\operatorname{tr}(S_t\Sigma_t^{-1}\mathbf{x}\mathbf{x}^\top) \\
&= -2\operatorname{tr}(S_t(I - \Sigma_t^{-1}\mathbf{x}\mathbf{x}^\top))\rho_t.
\end{aligned} \tag{52}$$

Note $\nabla \Phi_t(\mathbf{x}) = 2S_t\mathbf{x}$. Expanding the divergence yields

$$\begin{aligned}
-\nabla \cdot (\rho_t \nabla \Phi_t(\mathbf{x})) &= -2\sum_{i=1}^{d}\partial_i(\rho_t(\mathbf{x})(S_t\mathbf{x})_i) = -2\sum_{i=1}^{d}\left[(\partial_i\rho_t)(S_t\mathbf{x})_i + \rho_t(\mathbf{x})(S_t)_{ii}\right] \\
&= -2\sum_{i=1}^{d}\left[-(\Sigma_t^{-1}\mathbf{x})_i(S_t\mathbf{x})_i + (S_t)_{ii}\right]\rho_t(\mathbf{x}) \\
&= -2\rho_t(\mathbf{x})\left[-\mathbf{x}^\top\Sigma_t^{-1}S_t\mathbf{x} + \operatorname{tr}(S_t)\right] \\
&= -2\operatorname{tr}(S_t(I - \Sigma_t^{-1}\mathbf{x}\mathbf{x}^\top))\rho_t(\mathbf{x}).
\end{aligned} \tag{53}$$

Comparing (52) and (53) shows that

$$\dot{\rho}_t = -\nabla \cdot (\rho_t \nabla \Phi_t),$$

thereby establishing the continuity equation.

**Projection equation**: $\Phi_t + \operatorname{proj}_{\mathcal{C}(\rho_t),\rho_t}[0] = 0$.

By definition, we obtain

$$\operatorname{proj}_{\mathcal{C}(\rho_t),\rho_t}[0] = \underset{h \in \mathcal{C}(\rho_t)}{\arg\min}\int \|\nabla h\|_2^2 d\rho_t,$$

where $\mathcal{C}(\rho_t) = \operatorname{conv}\left(\{\delta_{\rho_t} F_k(\rho_t) : k = 1, 2, \dots, K\}\right)$.

Using (45) and (46), any function $h \in \mathcal{C}(\rho_t)$ satisfies

$$\nabla h(\mathbf{x}) = \sum_{k=1}^{K} w_k\left[\left(\Sigma_*^k\right)^{-1} - \Sigma_t^{-1}\right]\mathbf{x} = 2\sum_{k=1}^{K} w_k \nabla F_k^\dagger(\Sigma_t)\mathbf{x},$$

for some $\mathbf{w} \in \mathcal{W}$.

Substituting into the projection integral and using $E_{\rho_t}\left[\mathbf{x}^\top \mathbf{A}\mathbf{x}\right] = \mathrm{tr}(\mathbf{A}\Sigma_t)$ yields

$$\int \|\nabla h\|_2^2 d\rho_t = 4 \left\| \sum_{k=1}^{K} w_k \nabla F_k^\dagger(\Sigma_t) \right\|_{\Sigma_t}^2.$$

Since $4 > 0$ does not affect the minimizer, i.e.,

$$\mathbf{w}_t = \arg\min_{\mathbf{w}\in\mathcal{W}} \int \|\nabla h\|_2^2 d\rho_t = \arg\min_{\mathbf{w}\in\mathcal{W}} \left\| \sum_{k=1}^{K} w_k \nabla_{\Sigma_t} F_k^\dagger(\Sigma_t) \right\|_{\Sigma_t}^2.$$

As $\mathbf{w}_t$ is the optimal solution of the projection problem, the corresponding projected function is

$$h^*(\mathbf{x}) = \sum_{k=1}^{K} w_{t,k} \delta_{\rho_t} F_k(\rho_t)(\mathbf{x}) = -\mathbf{x}^\top S_t \mathbf{x} + C(t),$$

where $C(t)$ is a constant with respect to $\mathbf{x}$. Therefore,

$$\Phi_t(\mathbf{x}) = -\mathrm{proj}_{\mathcal{C}(\rho_t),\rho_t}[0],$$

which establishes the projection equation.

It remains to show that $\Sigma_t$ is positive definite for all $t \geq 0$.

**Positive definiteness property**: $\Sigma_t \succ 0$, for all $t \geq 0$.

Define the maximal time of positive definiteness: $T^* = \sup\{T > 0 : \Sigma_t \succ 0 \text{ for all } t \in [0,T)\}$. Let $t \in [0, T^*)$. As shown above, the MWGraD flow (9) and the covariance dynamics (11) are equivalent since $\Sigma_t \succ 0$.

In the proof of Lemma B.2, we show that the quantity $E_k(\rho_t, q) = F_k(\rho_t) - F_k(q)$ is non-increasing in $t$ for every $k \in [K]$ and any distribution $q$. Replacing $q$ by $\pi_k$, we obtain $E_k(\rho_t, \pi_k) = F_k(\rho_t)$. Then, it follows that

$$F_k^\dagger(\Sigma_t) = F_k(\rho_t) \leq F_k(\rho_0) = F_k^\dagger(\Sigma_0).$$

We can rewrite $F_k^\dagger(\Sigma_t)$ as follows

$$F_k^\dagger(\Sigma_t) = \frac{1}{2}\left[\mathrm{tr}(\Sigma_t(\Sigma_*^k)^{-1}) - \log\det\left(\Sigma_t(\Sigma_*^k)^{-1}\right) - d\right] = \frac{1}{2}\sum_{i=1}^{d}\ell(\mu_i),$$

where $\mu_1, \mu_2, \ldots, \mu_d$ are the eigenvalues of $\Sigma_t(\Sigma_*^k)^{-1}$ and $\ell(\mu) = \mu - \log\mu - 1 \geq 0$. Let $\mu_{\min} = \min_{i\in[d]}\mu_i$, we get

$$0 \leq \frac{1}{2}\ell(\mu_{\min}) \leq \frac{1}{2}\sum_{i=1}^{d}\ell(\mu_i) \leq F_k^\dagger(\Sigma_0).$$

Since $\ell(\mu) \to +\infty$ as $\mu \to 0$, the bound

$$\ell(\mu_{\min}) \leq 2F_k^\dagger(\Sigma_0)$$

implies the existence of a constant $c_1 > 0$ such that $\mu_{\min} \geq c_1$.

By Lemma D.1, $\Sigma_t(\Sigma_*^k)^{-1}$ shares its eigenvalues with the symmetric matrix $\Sigma_t^{1/2}(\Sigma_*^k)^{-1}\Sigma_t^{1/2}$. Using $(\Sigma_*^k)^{-1} \preceq \|(\Sigma_*^k)^{-1}\|_{\mathrm{op}}\mathbf{I}_d$, we have

$$\mu_{\min} = \min_{\|\mathbf{v}\|_2=1} \mathbf{v}^\top \Sigma_t^{1/2}(\Sigma_*^k)^{-1}\Sigma_t^{1/2}\mathbf{v} \leq \|(\Sigma_*^k)^{-1}\|_{\mathrm{op}} \min_{\|\mathbf{v}\|_2=1} \mathbf{v}^\top \Sigma_t \mathbf{v} = \|(\Sigma_*^k)^{-1}\|_{\mathrm{op}}\lambda_{\min}(\Sigma_t).$$

Therefore,

$$\lambda_{\min}(\Sigma_t) \geq \frac{\mu_{\min}}{\|(\Sigma_*^k)^{-1}\|_{\mathrm{op}}} \geq \frac{c_1}{\|(\Sigma_*^k)^{-1}\|_{\mathrm{op}}} =: c > 0.$$

Suppose $T^* < \infty$. By continuity of eigenvalues (Lemma D.2),

$$\lambda_{\min}(\Sigma_{T^*}) = \lim_{t \to T^*} \lambda_{\min}(\Sigma_t) \geq c > 0.$$

So $\Sigma_{T^*} \succ 0$, which contradicts the maximality of $T^*$. Therefore $T^* = \infty$, hence $\Sigma_t \succ 0$ for all $t \geq 0$.

$\square$

*Proof.* (of Theorem 3.6) Assuming $\Sigma_t \succ 0$, we establish the equivalence between the A-MWGraD flow (14) and the covariance dynamics (15).
**Continuity Equation**: $\dot{\rho}_t + \nabla \cdot (\rho_t \nabla \Phi_t) = 0$.
Note that $\dot{\Sigma}_t = 2(S_t \Sigma_t + \Sigma_t S_t)$ and $\nabla \Phi_t(\mathbf{x}) = 2S_t \mathbf{x}$. Repeating the derivation in (52)–(53), we obtain $\dot{\rho}_t = -\nabla \cdot (\rho_t \nabla \Phi_t)$.
**Hamilton-Jacobi Equation**: $\dot{\Phi}_t + \alpha_t \Phi_t + \frac{1}{2}\|\nabla \Phi_t\|_2^2 + \mathrm{proj}_{\mathcal{C}(\rho_t),\rho_t}[0] = 0$.

Since $\dot{\Phi}_t = \mathbf{x}^\top \dot{S}_t \mathbf{x} + \dot{C}_t$, and using $\dot{S}_t + \alpha_t S_t + 2S_t^2 = -\sum_{k=1}^K w_{t,k} \nabla_{\Sigma_t} F_k^\dagger(\Sigma_t)$, $\|\nabla \Phi_t(\mathbf{x})\|_2^2 = \|2S_t \mathbf{x}\|_2^2 = 4\mathbf{x}^\top S_t^2 \mathbf{x}$:

$$
\begin{aligned}
\dot{\Phi}_t + \alpha_t \Phi_t + \frac{1}{2}\|\nabla \Phi_t\|_2^2 &= \mathbf{x}^\top \dot{S}_t \mathbf{x} + \alpha_t \mathbf{x}^\top S_t \mathbf{x} + \dot{C}_t + \alpha_t C(t) + 2\mathbf{x}^\top S_t^2 \mathbf{x} \\
&= \mathbf{x}^\top \left[\dot{S}_t + \alpha_t S_t + 2S_t^2\right]\mathbf{x} + \dot{C}_t + \alpha_t C(t) \\
&= \mathbf{x}^\top \left[-\sum_{k=1}^K w_{t,k} \nabla_{\Sigma_t} F_k^\dagger(\Sigma_t)\right]\mathbf{x} - 1 + \frac{1}{2}\sum_{k=1}^K w_{t,k}\log\det\left(\Sigma_t(\Sigma_*^k)^{-1}\right) + \alpha_t C(t) \\
&= \sum_{k=1}^K w_{t,k}\left[-\mathbf{x}^\top \frac{1}{2}\left((\Sigma_*^k)^{-1} - \Sigma^{-1}\right)\mathbf{x} - \frac{1}{2}\log\frac{\det\Sigma_*^k}{\det\Sigma_t} - 1\right] + \alpha_t C(t) \\
&= -\sum_{k=1}^K w_{t,k}\delta_{\rho_t} F_k(\rho_t) + \alpha_t C(t),
\end{aligned}
$$

which means that $-\left(\dot{\Phi}_t + \alpha_t \Phi_t + \frac{1}{2}\|\nabla \Phi_t\|_2^2\right) \in \mathcal{C}(\rho_t)$. Furthermore, as shown in the proof of Theorem 3.1,

$$\mathbf{w}_t = \arg\min_{\mathbf{w} \in \mathcal{W}} \left\|\sum_{k=1}^K w_k \nabla_{\Sigma_t} F_k^\dagger(\Sigma_t)\right\|_{\Sigma_t}^2 = \arg\min_{\mathbf{w} \in \mathcal{W}} \int \|\nabla \delta_{\rho_t} F_k(\rho_t)\|_2^2 \, d\rho_t,$$

which means that the second equation holds.

**Positive definiteness property**: $\Sigma_t \succ 0$, for all $t \geq 0$.

Similarly, we define $T^* = \sup\{T > 0 : \Sigma_t \succ 0 \text{ for all } t \in [0, T)\}$, and prove that $T^* = \infty$. Suppose $t \in [0, T^*)$. As shown above, the A-MWGraD flow (14) is equivalent to the covariance dynamics (15) whenever $\Sigma_t \succ 0$. Furthermore, The corollary C.2 shows that $F_k(\rho_t) \leq F_k(\rho_0)$ for $k \in [K]$, implying that $F_k^\dagger(\Sigma_t)$ is upper bounded by a constant

$$F_k^\dagger(\Sigma_t) = F_k(\rho_t) \leq W_k(\rho_t) \leq W_k(\rho_0).$$

The remainder of the argument is identical to that used in the proof of Theorem 3.1. In particular, $\lambda_{\min}(\Sigma_t) > 0$ on $[0, T^*)$, which implies $T^* = \infty$. Therefore, $\Sigma_t \succ 0$ for all $t \geq 0$.

$\square$

## E. Detailed Derivation of (18)

Starting from (14), we have

$$\dot{\rho}_t + \nabla \cdot (\rho_t \nabla \Phi_t) = 0,$$

which is the continuity equation of $\rho_t$. At the particle level, each particle $\mathbf{x}_t$ evolves according to

$$\dot{\mathbf{x}}_t = \nabla \Phi_t(\mathbf{x}_t).$$

Let define the velocity $\mathbf{v}_t = \nabla \Phi_t(\mathbf{x}_t)$, its time derivative is

$$\begin{aligned}
\dot{\mathbf{v}}_t &= \partial_t \nabla \Phi_t(\mathbf{x}_t) + [\nabla^2 \Phi_t(\mathbf{x}_t)] \nabla \Phi_t(\mathbf{x}_t) \\
&= \left( -\alpha_t \nabla \Phi_t(\mathbf{x}_t) - [\nabla^2 \Phi_t(\mathbf{x}_t)] \nabla \Phi_t(\mathbf{x}_t) - \sum_{k=1}^{K} w_{t,k} \nabla \delta_\rho F_k(\rho_t)(\mathbf{x}_t) \right) + [\nabla^2 \Phi_t(\mathbf{x}_t)] \nabla \Phi_t(\mathbf{x}_t) \\
&= -\alpha_t \mathbf{v}_t - \sum_{k=1}^{K} w_{t,k} \nabla \delta_\rho F_k(\rho_t)(\mathbf{x}_t).
\end{aligned}$$

This gives the particle-level evolution equation (18) as desired.

## F. Algorithms: MWGraD and A-MWGraD

---

**Algorithm 1** Multi-objective Wasserstein Gradient Descent (**MWGraD**) (Nguyen et al., 2025a)

---

**Input:** Functionals $\{F_k\}_{k=1}^K$, number of particles $m$, number of iterations $T$, step sizes $\eta > 0$.
**Output:** a set of $m$ particles $\left\{\mathbf{x}_i^{(T)}\right\}_{i=1}^m$.
Sample $m$ initial particles $\left\{\mathbf{x}_i^{(0)}\right\}_{i=1}^m$ from $\mathcal{N}(0, \mathbf{I}_d)$.
  $n \leftarrow 0$
  **while** $n < T$ **do**
    Estimate $\bar{\Delta}_k^{(n)}(\mathbf{x}_i^{(n)})$ by (20) or (21), for $i \in [m]$, $k \in [K]$
    Compute $\mathbf{w}_n$ by solving (22)
    Update $\mathbf{x}_i^{(n+1)} \leftarrow \mathbf{x}_i^{(n)} - \eta \sum_{k=1}^K w_{n,k} \bar{\Delta}_k^{(n)}(\mathbf{x}_i^{(n)})$, for $i \in [m]$
    $n \leftarrow n + 1$
  **end**

---

---

**Algorithm 2** Accelerated Multi-objective Wasserstein Gradient Descent (**A-MWGraD**) (**This work**)

---

**Input:** Functionals $\{F_k\}_{k=1}^K$, number of particles $m$, number of iterations $T$, step sizes $\eta > 0$.
**Output:** a set of $m$ particles $\left\{\mathbf{x}_i^{(T)}\right\}_{i=1}^m$.
Sample $m$ initial particles $\mathbf{x}_i^{(0)}$ from $\mathcal{N}(0, \mathbf{I}_d)$ and set initial velocities as $\mathbf{v}_i^{(0)} = 0$, for $i \in [m]$.
$n \leftarrow 0$
  **while** $n < T$ **do**
    Estimate $\bar{\Delta}_k^{(n)}(\mathbf{x}_i^{(n)})$ by (20) or (21), for $i \in [m]$, $k \in [K]$.
    Compute $\mathbf{w}_n$ by solving (22).
    Set $\alpha_n = \begin{cases} \frac{1 - \sqrt{\beta\eta}}{1 + \sqrt{\beta\eta}}, & \text{if } F_k \text{ are } \beta\text{-strongly geodesically convex, for all } k \in [K] \\ \frac{n-1}{n+2}, & \text{if } F_k \text{ are geodesically convex or } \beta \text{ is unknown, for all } k \in [K]. \end{cases}$
    Update $\mathbf{x}_i^{(n+1)} \leftarrow \mathbf{x}_i^{(n)} + \sqrt{\eta} \mathbf{v}_i^{(n)}$.
    Update $\mathbf{v}_i^{(n+1)} \leftarrow \alpha_n \mathbf{v}_i^{(n)} - \sqrt{\eta} \sum_{k=1}^K w_{n,k} \bar{\Delta}_k^{(n)}(\mathbf{x}_i^{(n)})$, for $i \in [m]$.
    $n \leftarrow n + 1$.
  **end**

---

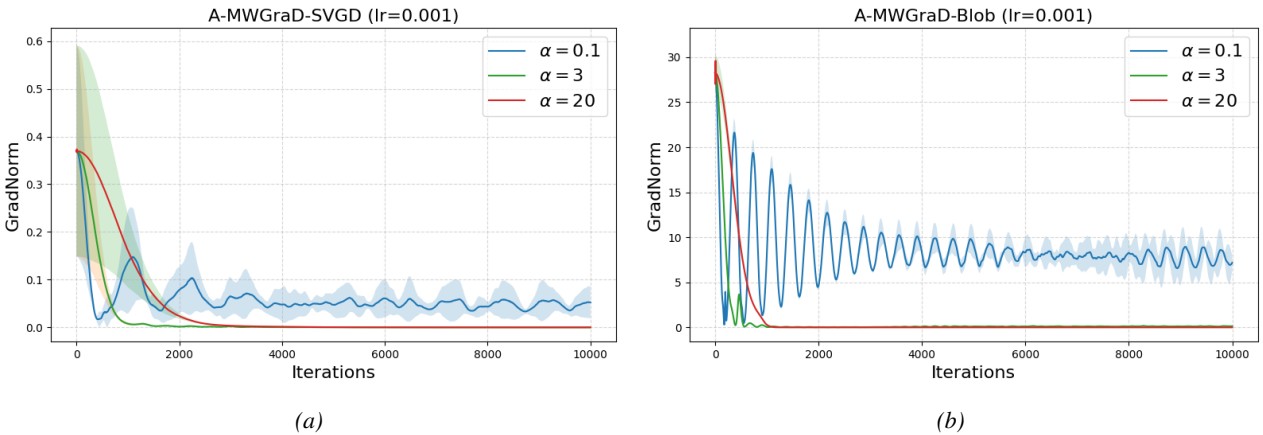

*Figure 3.* Convergence comparison: (a) A-MWGraD-SVGD, (b) A-MWGraD-Blob in terms of momentum parameter $\alpha$. The plot show mean and standard deviation (averaged over 5 trials) of $\mathrm{GradNorm}$ over 10000 iterations with three different momentum parameters $\alpha$ of $0.1, 3, 20$. The step size $\eta$ is fixed as $0.001$.

## G. Additional Experimental Results

We conduct ablation studies to evaluate the sensitivity of A-MWGraD to the momentum parameter, kernel bandwidth, particle count, and the cost of combining multiple Wasserstein gradients.

### G.1. Momentum parameter $\alpha$

First, we investigate the effect of the momentum parameter (or damping coefficient) $\alpha$ (see Theorem 3.7) on the convergence behavior and stability of the proposed accelerated algorithms.

We consider the multi-target sampling problem involving four target distributions $\pi_k = \mathcal{N}(\mu_k, \Sigma_k)$ $(k \in [4])$, where each target is Gaussian. Consequently, the objective functions $F_k(\rho) = KL(\rho, \pi_k)$ are geodesically convex. The parameters of the target distributions are given by

$$\mu_1 = [2.83, -2.74], \Sigma_1 = [[4.19, -3.44], [-3.44, 4.70]], \mu_2 = [-2.83, 2.89], \Sigma_2 = [[4.19, -3.03], [-3.03, 3.87]],$$
$$\mu_3 = [-2.68, -2.92], \Sigma_3 = [[5.07, 3.33], [3.33, 3.72]], \mu_4 = [2.74, 2.89], \Sigma_4 = [[4.70, 3.26], [3.26, 3.87]].$$

The distribution $\rho$ is represented by 50 particles initialized from a standard Gaussian distribution. We update particles using A-MWGraD-SVGD and A-MWGraD-Blob. For both methods, the momentum parameter is set as $\alpha_n = (n+2-\alpha)/(n+2)$, which corresponds to the continuous-time damping coefficient $\alpha_t = \alpha/t$ in Theorem 3.7. We measure the squared norm of the combined Wasserstein gradients (23), whose vanishing characterizes convergence to a weakly Pareto-optimal solution.

We consider $\alpha \in \{0.1, 0.5, 3.0, 10, 20\}$ and plot the evolution of this quantity over 10,000 iterations in Figure 3. For clarity, we show the representative cases $\alpha = 0.1$, $\alpha = 3$, and $\alpha = 20$. The results reveal a dependence on the momentum parameter. For small values of $\alpha$ (e.g., $\alpha = 0.1$), we observe large oscillations near the optimum, where the squared norm is close to zero, and repeatedly overshoot the solution. In contrast, when $\alpha$ is large (e.g., $\alpha = 20$), A-MWGraD behaves similarly to the (unaccelerated) MWGraD, and the acceleration effect largely disappears. Among the tested values, $\alpha = 3$ provides the fastest convergence.

These observations are consistent with our continuous-time analysis. In the proof of Theorem 3.7 (see Appendix C), we establish that $\dot{\mathcal{E}}_{k,\lambda}(\rho_t, q) \le t(3 - \alpha) \int \|\nabla \Phi_t\|_2^2 d\rho_t$. Therefore, to guarantee $\dot{\mathcal{E}}_{k,\lambda}(\rho_t, q) \le 0$, it is sufficient to require $\alpha \ge 3$. This theoretical result provides explains the following interpretation of the empirical behavior:

- $\alpha < 3$: $\mathcal{E}_{k,\lambda}$ may increase over time, resulting in oscillations and overshooting.

- $\alpha = 3$: the bound becomes tight, yielding the strongest acceleration predicted by the theory.

- $\alpha > 3$: stronger damping weakens the momentum effect, so the trajectory behaves closer to the MWGraD flow.

We further repeat the same study on the toy examples considered in Section 4, where the targets are Gaussian mixtures. Similar qualitative behavior is observed, despite the fact that the corresponding objective functions are generally not geodesically convex and therefore fall outside the scope of our theoretical analysis. Based on these observations, we set $\alpha = 3$ in the remaining experiments.

### G.2. Kernel bandwidth

We evaluate the RBF kernel with bandwidth $\sigma \in \{0.1, 1.0, 10.0, 100.0\}$ on Multi-MNIST and Multi-Fashion, reporting the average testing accuracies across two tasks.

*Table 2.* Multi-MNIST: Effect of kernel bandwidth $\sigma$

| $\sigma$ | 0.1 | 1 | 10 | 100 |
|---|---|---|---|---|
| A-MWGraD-SVGD | 92.8 | 94.3 | 93.9 | 93.3 |
| A-MWGraD-Blob | 92.5 | 94.2 | 94.3 | 93.2 |

*Table 3.* Multi-Fashion: Effect of kernel bandwidth $\sigma$

| $\sigma$ | 0.1 | 1 | 10 | 100 |
|---|---|---|---|---|
| A-MWGraD-SVGD | 85.7 | 86.3 | 86.1 | 85.2 |
| A-MWGraD-Blob | 84.9 | 86.4 | 86.6 | 85.1 |

As shown in Tables 2 and 3, performance is stable for $\sigma = 1$ and 10, while very small or large bandwidths ($\sigma = 0.1$ or 100) slightly degrade accuracy.

### G.3. Particle count

We vary the number of particles $K \in \{2, 5, 10, 20\}$.

*Table 4.* Multi-MNIST: Effect of particle count $K$

| $K$ | 2 | 5 | 10 | 20 |
|---|---|---|---|---|
| A-MWGraD-SVGD | 92.8 | 94.3 | 94.1 | 94.5 |
| A-MWGraD-Blob | 92.2 | 94.2 | 94.6 | 94.1 |

*Table 5.* Multi-Fashion: Effect of particle count $K$

| $K$ | 2 | 5 | 10 | 20 |
|---|---|---|---|---|
| A-MWGraD-SVGD | 84.6 | 86.3 | 86.2 | 86.1 |
| A-MWGraD-Blob | 85.1 | 86.4 | 86.2 | 86.6 |

When $K = 2$, the performance decreases. Increasing beyond 5 provides limited improvement while significantly increasing training cost, so $K = 5$ offers a good trade-off.

### G.4. Number of objectives

Both MWGraD and A-MWGraD require solving for $\mathbf{w}$ to combine multiple Wasserstein gradients at each iteration. We evaluate the computational overhead for solving $\mathbf{w}$. We conduct an ablation with increasing numbers of objectives $K \in \{2, 4, 10, 20\}$. We measure the ratio between the runtime required to solve for $\mathbf{w}$ and the total runtime of one iteration. The results, averaged over 5 trials, are shown in Table 6.

*Table 6.* Ratio of $w$-solving runtime to total iteration runtime, averaged over 5 trials.

| $K$ | 2 | 4 | 10 | 20 |
|---|---|---|---|---|
| MWGraD-SVGD | 0.1238 | 0.4379 | 0.7664 | 0.7894 |
| A-MWGraD-SVGD | 0.0728 | 0.3636 | 0.6907 | 0.6842 |
| MWGraD-Blob | 0.1168 | 0.3812 | 0.4551 | 0.6812 |
| A-MWGraD-Blob | 0.0798 | 0.3703 | 0.4457 | 0.6724 |

We observe that this ratio increases as the number of objectives grows. This indicates that when $K$ is large, the cost of solving for $\mathbf{w}$ can become a dominant component of each iteration, and more efficient strategies for computing $\mathbf{w}$ is desirable. In fact, the original MWGraD (Nguyen et al., 2025a) proposes updating $\mathbf{w}$ iteratively rather than solving the optimal one at each iteration, which can reduce the computational cost when $K$ is large. Considering such strategies is one future work when we have problems with large $K$ objectives.

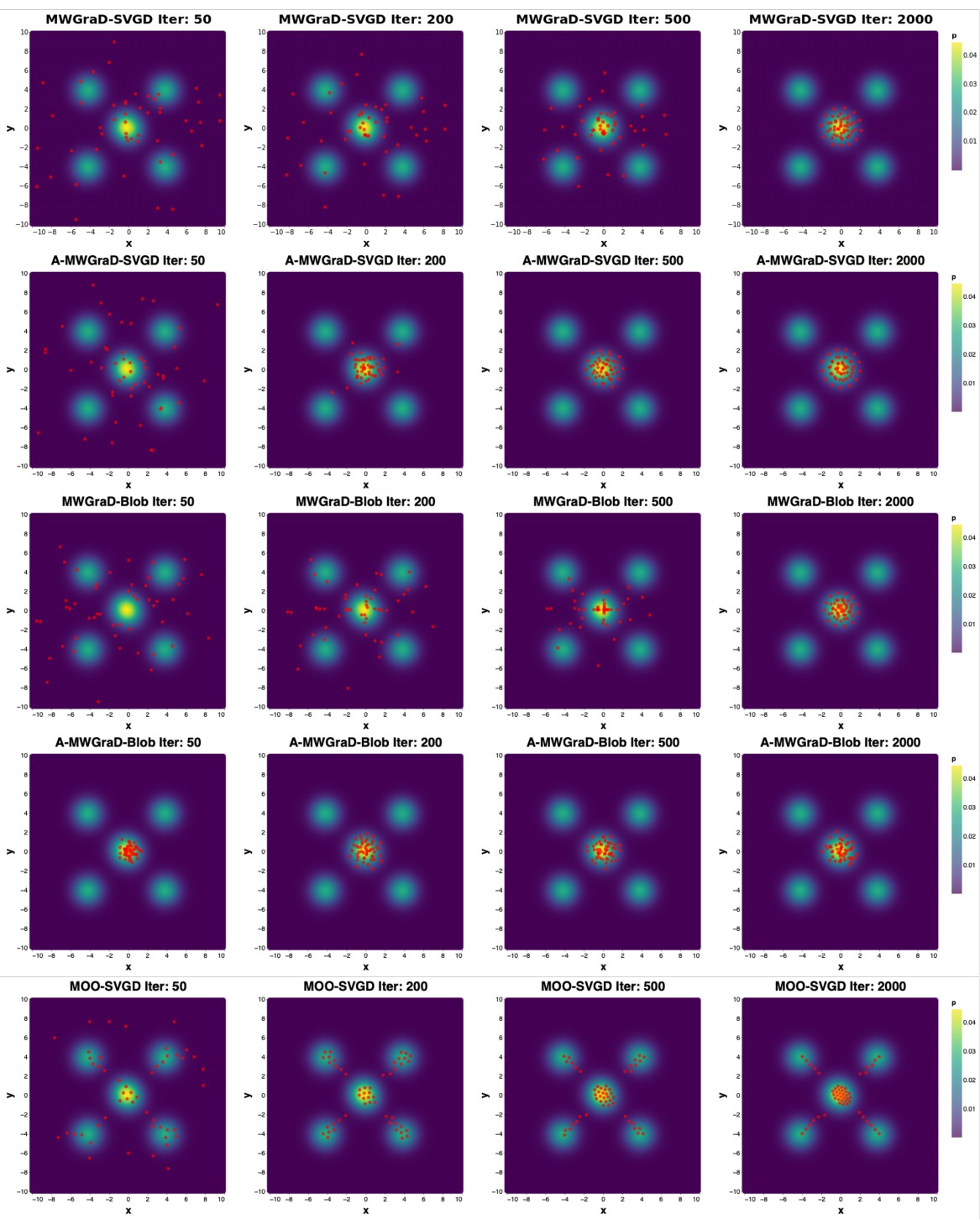

*Figure 4.* Sampling from multiple target distributions, where each target is a mixture of two Gaussians. These targets have a joint high-density region around the origin. Initially, 50 particles are sampled from the standard distribution, and then updated by MWGraD variants, A-MWGraD variants, and MOO-SVGD. While MOO-SVGD tends to scatter particles across all the modes (see the last row), MWGraD and A-MWGraD variants tend to move particles towards the joint high-density region. Furthermore, A-MWGraD variants converge faster than MWGraD variants.

