# OpenReview forum: "Accelerated Multiple Wasserstein Gradient Flows for Multi-objective Distributional Optimization"
_ICML.cc/2026/Conference — ICML 2026 regular_

### Official Review · Reviewer_ULBH · 2026-02-15

**Soundness:** 2
**Presentation:** 3
**Significance:** 3
**Originality:** 3
**Overall Recommendation:** 3
**Confidence:** 4

**Summary:**

This paper generalizes a recently introduced Wasserstein gradient descent algorithm for multi-objective optimization. The authors propose an accelerated variant in the spirit of Nesterov's acceleration. They prove that the flow's convergence rate matches the known rate from finite-dimensional optimization under geodesic convexity of the objective. The Lyapunov functions used in the convergence analysis (merit functions in the paper's terminology) are constructed by analogously following their Euclidean counterparts. Numerical implementation of the flow is discussed and several experiments which support the theoretical findings are provided.

**Compliance With Llm Reviewing Policy:**

Affirmed.

**Final Justification:**

I thank the authors for their engagement with my questions. My main concern remains the mathematical rigor of the paper. While I understand that the well-posedness of the proposed flows is not the central focus, and that partial results for the Gaussian family, as provided in the rebuttal, may be sufficient in that regard, the assumptions concerning the existence and differentiability of optimal transport maps are more serious. These assumptions form the backbone of the convergence analysis, which is itself the main focus of the paper. In this respect, I echo reviewer uTT5’s comments regarding the regularity of optimal transport maps. Concerning the existence of such maps, I maintain the point raised in my initial assessment. Their existence is assumed only along the flow, that is, for each $t$, there exists an optimal transport map $T_t$ from $\rho_t$ to, say, $q$, but this is not established beyond the Gaussian case discussed in the rebuttal. In my view, given the complexity of the flows under consideration, proving this in general is both highly nontrivial and necessary in order to support a general convergence result beyond the Gaussian setting. In light of these limitations, I have decided to maintain my initial score.

**Key Questions For Authors:**

Please see Weaknesses section.

**Limitations:**

Yes

**Strengths And Weaknesses:**

### Strengths
* The paper is clearly written and well structured. I could follow the story of the paper and I particularly appreciate that the authors provided proof sketches in the main body.
* Another good point is the brief presentation on gradient and Hamiltonian flows on Euclidean and probability spaces which improves readability.
* The discussion on particles implementation and numerical experiments is detailed and clearly support the theoretical claims.
* I appreciate the authors highlighting in the conclusions section that the discrete time convergence analysis of these kind of accelerated Wasserstein gradient flows remains open since I also agree it is a challenging problem but nevertheless the authors made a promising step by providing the continuous time analysis.

### Weaknesses
My comments are mainly technical:
* In Theorems 3.4 and 3.5 one of the key assumptions is that flows (9) and (13), respectively, are well-posed, i.e., that $(\rho_t)_t$ is a solution to (9) and (13), respectively. I understand well-posedness of the flows is not the main focus of the paper but I believe it is still important that the authors provide at least a proof sketch for that or point to some references in the existing literature that those flows admit solution. For example, as the authors claim at the beginning of Section 3, for (9) it seems that the classical approach of sending stepsize to zero in the discrete-time scheme should in principle be expected to work. Nevertheless, since (9) and (13) are quite complicated flows it would be good to see more details.

* The notion of geodesic convexity you use in Assumption 3.3 implicitly assumes the existence of an optimal transport map between $p$ and $q$. I checked the proofs of Theorems 3.4 and 3.5 and in order to apply Assumption 3.3 you implicitly assume that along the flow, i.e., for each $t$, there exists an optimal transport map $T_t$ from $\rho_t$ to, say, $q$. Such a map doesn't always exist and a sufficient condition given by Brenier's theorem is that the source measure is absolutely continuous w.r.t. Lebesgue measure. In your setting this amounts to proving that starting your flow from absolutely continuous $\rho_0$, absolute continuity propagates in time, i.e., for all $t > 0$, $(\rho_t)_{t >0}$, which solves the flow, remains absolutely continuous. Could the authors comment if they believe this could be proved? Otherwise, perhaps Assumption 3.3 can be reformulated in a weaker form because in its current state and how it is applied doesn't always hold. Also in this regard, another point is assuming differentiability of the OT map as in (23) and (24) for example. Regularity of OT maps is quite delicate so I believe extra assumptions would be needed.

* This is more of a general comment. It seems that the proof techniques are mainly adapted from the convergence analysis of accelerated methods in Euclidean spaces. I believe the paper would benefit from a more thorough discussion of the challenges involved in adapting these techniques to the Wasserstein space.

---

> ### Author Rebuttal · Authors · 2026-03-27
>
> We thank the reviewer for the constructive comments. We will address the points raised individually below.
>
> > In Theorems 3.4 and 3.5 one of the key assumptions is that flows (9) and (13), respectively, are well-posed, i.e., that $(\rho_t)_t$  is a solution to (9) and (13), respectively. I understand well-posedness of the flows is not the main focus of the paper but I believe it is still important that the authors provide at least a proof sketch for that or point to some references in the existing literature that those flows admit solution...
>
> As the reviewer mentioned, the main focus of this work is to design an accelerated version of MWGraD flow (9) and to establish its improved convergence properties in the continuous-time limit. Therefore, we do not attempt to prove the well-posedness of the flows i.e. the existence of the solutions of flows (9) and (13) *in full generality*. However, we agree that it is useful to prove some justification that the flows admit solutions in certain settings. To this end, we add an example showing that **when the distributions are restricted to Gaussian family with zero mean, the solutions of flows (9) and (13) indeed exist**. In the revision, we include the following propositions with their proofs.
>
>
> Consider the $k$-th objective $F_{k}(\rho)= KL(\rho, \pi_{k})$. Suppose that the initial distribution $\rho_{0}$ and the target $\pi_{k}$ are Gaussian with zero mean and covariance matrices $\Sigma_{0}$ and $\Sigma_{k}^{*}$, respectively, for $k\in [K]$.
>
> **Proposition 1** (MWGraD flow (9) in Gaussian family).
>
> Suppose $\rho_t=\mathcal{N}(0, \Sigma_t)$, where $\Sigma_t$ solves
>
> $\dot{\Sigma_{t}}=- 2(G_t \Sigma_t + \Sigma_t G_t)=0$,
>
> with the initial values $\Sigma_0$,  where
>
> $G_t=\sum_{k=1}^{K}w_{t,k} \nabla_{\Sigma_t} F_k(\rho_t)$.
>
> Then, for any $t\geq 0$, $\Sigma_t$ is positive definite.
>
> Furthermore, let $\phi_t(x)=-x^{T}G_t x$. Then $(\rho_t, \phi_t)$ is a solution to the MWGraD flow (9) with initial values $\rho_0$ and $\phi_0=0$.
>
> **Proposition 2** (A-MWGraD flow (13) in Gaussian family)
>
> Suppose $\rho_t=\mathcal{N}(0, \Sigma_t)$, and let $(\Sigma_t, S_t)$ solves
>
> $\dot{\Sigma_{t}}- 2(S_t \Sigma_t + \Sigma_t S_t)=0$
>
> $\dot{S_{t}}+\alpha_t S_t + 2 S_t^2 + \sum_{k=1}^{K}w_{t,k} \nabla_{\Sigma_t}F_k(\rho_t) =0$,
>
> with initial values $\Sigma_0$ and $S_0 =0$.
>
> Then, for any $t\geq 0$, $\Sigma_t$ is positive definite.
>
> Furthermore, let $\phi_t(x)=x^{T}S_t x+C(t)$,
>
> where $C(t)=-t + \frac{1}{2}\sum_{k=1}^{K}\int_0^{t} w_{s,k} \log\det (\Sigma_s (\Sigma_k^{*})^{-1})ds$.
>
> Then, $(\rho_t, \phi_t)$ is a solution to the A-MWGraD flow (13) with initial values $\rho_0$ and $\phi_0=0$.
>
> In these propositions, we **explicitly construct the solutions** of flows (9) and (13) when the distributions are restricted to the Gaussian family.
>
>
> > The notion of geodesic convexity you use in Assumption 3.3 implicitly assumes the existence of an optimal transport map between $p and q. I checked the proofs of Theorems 3.4 and 3.5 and in order to apply Assumption 3.3 you implicitly assume that along the flow, ...
>
> Following the reviewer's suggestion, we assume the existence of optimal transport map between $p$ and $q$ in Assumption 3.3. Furthermore, in (23) and (24), we assume differentiability of optimal transport maps. As discussed in the previous response, these assumption are satisfied when the distributions are restricted to the Gaussian family. We have revised manuscript accordingly.
>
>
> > This is more of a general comment. It seems that the proof techniques are mainly adapted from the convergence analysis of accelerated methods in Euclidean spaces. I believe the paper would benefit from a more thorough discussion of the challenges...
>
> We follow Lyapunov-function approach used for accelerated methods  in Euclidean spaces. Specifically, a similar approach was used by Li &Wang to analyze optimization over probability spaces with one single objective. However, extending this to multi-objective optimization in probability space has several challenges.
>
> First, while Li & Wang upper bound the $F(\rho_t)-F(\rho^{*})$, our goal is to bound the merit function $\max_q \min_{k\in[K]} [F_k(\rho)- F_k(q)]$. We address this by adapting the Lyapunov function by to control the inner min term, and then bounding the outer max using our Lemma 3.2 and lemma B.1.
>
> Second, unlike Li & Wang, we must ensure that $\rho_t$ remains in the level set $\Omega_{\textbf{F}}(\textbf{F}(\rho_0))$,  in order to apply Lemma 3.2. We prove Lemma B.2 and Lemma C.1 to guarantee this property for MWGraD (9) and A-MWGraD (13).
>
> In summary, adapting the proof techniques from single-objective to multi-objective optimization, and from Euclidean space to probability space, is **highly non-trivial**. We have added a discussion of these challenges in the revision.
>
> We'd like to thank the reviewer again for evaluating our paper. We hope that we have addressed your concerns. Please let us know if you have any more questions.

---

> > ### Author Rebuttal · Reviewer_ULBH · 2026-04-02
> >
> > Thank you to the authors for their detailed response. While some of my concerns have been partially addressed the main limitation remains: the paper’s technical framework is fully validated only in the Gaussian setting. I understand that well-posedness of the flows may not be central since the paper’s primary goal is convergence. However, for the convergence analysis itself, the assumptions that optimal transport maps exist and are differentiable appear to hold only in the Gaussian case, which significantly limits the generality of the results.
> >
> > This leads to the following follow-up question:
> >
> > Would it be possible to define geodesic convexity along optimal transport plans $\gamma^* \in \Gamma(p,q)$ which always exist for $W_2$, rather than along transport maps? At present, the paper assumes the transport plan is unique and induced by a map T, i.e., $\gamma^* =(Id, T){\\#}p$. Under this alternative definition, would it still be possible to establish the same convergence results?

---

> > > ### Author Response · Authors · 2026-04-02
> > >
> > > We thank the reviewer for the insightful question.
> > >
> > > Our convergence analysis relies on the existence and differentiability of the OT map. In our response, we showed that these assumptions are satisfied in the Gaussian setting considered in our analysis. We agree that defining geodesic convexity along OT plans rather than OT maps, and investigating whether similar convergence guarantees can be obtained, is an interesting direction. We will include this point as a potential direction for future work.

---

### Official Review · Reviewer_2Dba · 2026-03-11

**Soundness:** 3
**Presentation:** 4
**Significance:** 3
**Originality:** 3
**Overall Recommendation:** 5
**Confidence:** 4

**Summary:**

This paper proposes a pioneering acceleration scheme for **Multi-Objective Distributional Optimization (MODO)**. The current state-of-the-art, MWGraD, is limited by a convergence rate of $\mathcal{O}(1/t)$ in the continuous-time limit. Inspired by Nesterov’s acceleration and the perspective of **damped Hamiltonian dynamics**, the authors introduce **A-MWGraD**. The work establishes rigorous theoretical convergence rates in Wasserstein space and provides a practical particle-based implementation using kernel approximations. The results demonstrate a significant leap in both sampling efficiency and optimization speed.

**Compliance With Llm Reviewing Policy:**

Affirmed.

**Final Justification:**

I was thinking this is a good work and keep my rating.

**Key Questions For Authors:**

1. In the discrete implementation, does the momentum term ever cause "overshooting" on the Pareto front, and how sensitive is the algorithm to the choice of the damping parameter $\alpha$?

2. For the number of targets $K$ being large, would the computational overhead of solving for $w$ offset the advantage of acceleration?

**Limitations:**

yes

**Strengths And Weaknesses:**

- **Strengths**:

    - **Groundbreaking Theory**: Addressing the acceleration of MODO is a non-trivial and highly original contribution.

    - **Practical Efficiency**: The drastic reduction in iterations (e.g., from 500 to 50) is a massive gain for resource-intensive sampling tasks.

    - **Clarity**: The transition from physical intuition (Hamiltonian) to rigorous math is executed with great clarity.

- **Weaknesses**:

    - The gap between continuous-time theory and discrete-time implementation is acknowledged but not theoretically bridged.

    - The computational overhead of the per-iteration QP solver for large $K$ is not extensively discussed.

---

> ### Author Rebuttal · Authors · 2026-03-28
>
> We thank the reviewer for the constructive comments. We will address the points raised individually below.
>
> > In the discrete implementation, does the momentum term ever cause "overshooting" on the Pareto front, and how sensitive is the algorithm to the choice of the damping parameter $\alpha$?
>
> In our experiments, we conducted an ablation study to investigate how the damping parameter $\alpha$ affects the convergence and stability. In particular, we repeated the toy experiments in Subsection 4.1. We measure the squared norm of the combined Wasserstein gradients (Eq. 19), which characterizes the convergence to (weakly) Pareto-optimal solutions. We considered $\alpha\in 0.1, 1, 3, 5, 10$, and visualize the evolution of this quantity over 10,000 iterations, similar to Figure 1.
>
> Our observations are as follows. For small values ($\alpha=0.1, 1$), we observe **large oscillations around the optimum** (where the squared norm is near 0), and the trajectory **overshoots** multiple times. In contrast, for larger ($\alpha=5, 10$), the accelerated version A-MWGraD behaves similarly to the unaccelerated MWGraD, and the **acceleration effect largely disappears**. Among test values, $\alpha=3$ gives the **best acceleration**.
>
> These observations are consistent with our continuous-time analysis. In the proof of Theorem 3.5 (see Appendix C), we show that:
>
> $\dot{\mathcal{E}}_{\lambda}(\rho_t, q)\leq t(3-\alpha) \int |\nabla \phi|^2 d\rho_t$.
>
> To guarantee $\dot{\mathcal{E}}_{\lambda}(\rho_t, q)\leq 0$, we require $\alpha\geq 3$. This explains the empirical behavior:
>
> - $\alpha<3$: $\mathcal{E}_{\lambda}$ may increase, resulting in oscillations and overshooting.
> - $\alpha=3$: the bound becomes tight, yielding the strongest acceleration predicted by our analysis.
> - $\alpha >3$: stronger damping weakens the momentum effect, so the trajectory behaves closer to the unaccelerated gradient flow.
>
>
> In the revision, **we have included this ablation study together with the discussion above** of the role of $\alpha$.
>
> > For the number of targets $K$  being large, would the computational overhead of solving for  w offset the advantage of acceleration?
>
> We would like to clarify that the goal of this work is to **accelerate the convergence of the MWGraD flow, rather than to improve its runtime per iteration**. Furthermore, both MWGraD and our accelerated MWGraD require solving for w to combine multiple gradients at each iteration. Thus, the computational overhead for solving w is **not specific to our accelerated method** and is not primary focus of this work.
>
> To better understand this overhead, we conducted an ablation with increasing numbers of objectives $K\in [2,4,10,20]$. We measured the ratio between the runtime required to solve for w and the total runtime of one iteration. The results, averaged over 5 trials, are shown in the table below.
>
> | K | 2 | 4 | 10 | 20|
> |-----------|-----------|-----------|-----------|-----------|
> | MWGraD-SVGD | 0.1238 | 0.4379 | 0.7664 | 0.7894|
> | A-MWGraD-SVGD | 0.0728 | 0.3636 | 0.6907 | 0.6842|
> | MWGraD-Blob | 0.1168 | 0.3812 | 0.4551 | 0.6812|
> | A-MWGraD-Blob | 0.0798 | 0.3703 | 0.4457 | 0.6724|
>
>
> We observe that this ratio increases as the number of objectives grows. This indicates that when K is large, the cost of solving for w can become a dominant component of each iteration, and more efficient strategies for computing w is desirable. In fact, the original MWGraD (Nguyen et al) proposes updating w iteratively rather than solving the optimal one at each iteration, which can reduce the computational cost when K is large. Considering such strategies is one future work when we have problems with large K objectives. **We have included this ablation study into the revision**.
>
> We'd like to thank the reviewer again for evaluating our paper. We hope that we have addressed your concerns. Please let us know if you have any more questions.

---

> > ### Author Rebuttal · Reviewer_2Dba · 2026-04-03
> >
> > I have no more questions for this work

---

> > > ### Author Response · Authors · 2026-04-04
> > >
> > > We'd like to thank the reviewer again for evaluating our paper.

---

### Official Review · Reviewer_dJwV · 2026-03-13

**Soundness:** 3
**Presentation:** 3
**Significance:** 4
**Originality:** 3
**Overall Recommendation:** 5
**Confidence:** 3

**Summary:**

This paper studies multi-objective distributional optimization (MODO) in the Wasserstein space. Building on the recently proposed Multiple Wasserstein Gradient Descent (MWGraD) framework, the paper develops an accelerated counterpart, A-MWGraD, inspired by Nesterov-style acceleration and damped Hamiltonian dynamics in probability space.

The paper first introduces the continuous-time MWGraD flow (9) and shows that this flow achieves an $O(1/t)$ rate under geodesic convexity assumptions. It then proposes the accelerated flow A-MWGraD and proves faster continuous-time rates (Theorem 3.5): $O(1/t^2)$ for geodesically convex objectives and $O(e^{-\sqrt{\beta}t})$ for $\beta$-strongly geodesically convex objectives.

On the algorithmic side, the paper derives a discrete particle implementation using kernel approximations of Wasserstein gradients, with SVGD- and Blob-style variants given in (17)–(18). The experiments in Section 4 include both toy multi-target sampling and Bayesian multi-task learning benchmarks. Empirically, the accelerated variants are competitive or faster than their MWGraD counterparts.

**Compliance With Llm Reviewing Policy:**

Affirmed.

**Final Justification:**

The authors addressed my concerns from the review, I keep my score

**Key Questions For Authors:**

1) Can you better justify the connection between the continuous-time theory and the practical particle implementation? In particular, the main results are for exact flows, while the algorithm uses the kernel approximations in (17)-(18). Even a partial discrete-time argument, or a careful discussion of what aspects of the acceleration mechanism are expected to survive approximation, would strengthen the paper and could improve my soundness assessment.

2) How sensitive is A-MWGraD to the kernel bandwidth, particle count, and momentum schedule? Since the practical method depends on these choices, an ablation study would help establish robustness.

3) Can you clarify the baseline list in Section 4.2 and Table 1? The text mentions MT-SGD and later refers to MGDA, but neither appears clearly in Table 1.

4) Can you soften or better substantiate the claim that A-MWGraD variants "consistently outperform" other methods? As currently reported, Table 1 shows several strong results but not uniform domination.

**Limitations:**

yes

**Strengths And Weaknesses:**

### Strengths

1) The paper’s main strength is its continuous-time theory. It formulates the MWGraD flow (9), and proves faster rates for the accelerated flow in Theorems 3.4 and 3.5.

2) The contribution is conceptually interesting. Rather than adding momentum in an ad hoc way, the paper gives a principled accelerated extension of MWGraD in Wasserstein space, which makes the work feel both novel and theoretically grounded.

3) The problem is relevant, especially for multi-target sampling and Bayesian multi-task learning, so the paper has value beyond a purely theoretical exercise. The experiments support the claim that the accelerated variants can improve convergence and perform competitively in practice.

### Weaknesses

1) The main weakness is the gap between theory and implementation. The analysis is for continuous-time flows with exact Wasserstein gradients, while the practical method uses discrete particle updates and kernel approximations. This limits how directly the theory supports the algorithm that is actually tested.

2) The empirical claims are somewhat too strong. The paper says the accelerated methods “consistently outperform” others, but Table 1 is more mixed, with some tasks where non-accelerated MWGraD variants perform slightly better.

3) The experimental presentation also needs cleanup. The baseline description is inconsistent across the text and Table 1, and the reported number of runs differs between Figure 2 description in the text and Table 1.

---

> ### Author Rebuttal · Authors · 2026-03-29
>
> We thank the reviewer for the constructive comments. We will address the points raised individually below.
>
> > 1. Can you better justify the connection between the continuous-time theory and the practical particle implementation?...
>
> We focus on the continuous-time convergence analysis of the accelerated method A-MWGraD, assuming the exact flow can be computed, i.e. the Wasserstein gradient of objectives are available. In the revision, we show that when each distribution $\rho_t$ is assumed to be Gaussian, the Wasserstein gradients and flows can be exactly computed. However, the analysis for empirical particle approximation remains open. Nevertheless, establishing the continuous time analysis is an important step toward a complete theoretical understanding of accelerated methods for **multi-objective optimization over the probability space**.
>
> > 2. How sensitive is A-MWGraD to the kernel bandwidth, particle count, and momentum schedule? ...
>
> We conducted ablation studies to evaluate the sensitivity of A-MWGraD is to the momentum schedule, kernel bandwidth, and particle count.
>
> **Momentum schedule**
>
> We studied the damping parameter $\alpha=0.1, 1, 3, 5, 10$ using the toy experiments in Subsection 4.1. We measured the squared norm of the combined Wasserstein gradients (Eq. 19) over 10,000 iterations, which characterizes convergence to (weakly) Pareto-optimal solutions.
>
> For small values ($\alpha=0.1, 1$), the trajectory shows large **oscillations** around the optimum (where the squared norm is near 0), and overshoots multiple times. For larger values ($\alpha=5, 10$), A-MWGraD behaves similarly to the unaccelerated MWGraD, and the acceleration effect **weakens**. Among test values, $\alpha=3$ gives the best acceleration, consistent with our continuous-time analysis (see response to Reviewer 2Dba). We thus set $\alpha=3$ in the remaining experiments.
>
> **Kernel bandwidth**
>
> We evaluated the RBF kernel with bandwidth $\sigma=0.1, 1.0, 10.0,100.0$ on Multi-MNIST and Multi-Fashion, reporting the average testing accuracies across two tasks.
>
> **Multi-MNIST**
>
> | $\sigma$ | 0.1 | 1 | 10 | 100|
> |-----------|-----------|-----------|-----------|-----------|
> | A-MWGraD-SVGD | 92.8 | 94.3 | 93.9 | 93.3|
> | A-MWGraD-Blob | 92.5 | 94.2 | 94.3 | 93.2|
>
> **Multi-Fashion**
>
> | $\sigma$ | 0.1 | 1 | 10 | 100|
> |-----------|-----------|-----------|-----------|-----------|
> | A-MWGraD-SVGD | 85.7 | 86.3 | 86.1 | 85.2|
> | A-MWGraD-Blob | 84.9 | 86.4 | 86.6 | 85.1|
>
> Performance is stable for $\sigma=1$ and $10$, while very small or large bandwidths ($\sigma=0.1$ or $100$) slightly degrade accuracy.
>
> **Particle count**
>
> We also varied the number of particles $K=2,5,10, 20$.
>
> **Multi-MNIST**
>
> | K | 2 | 5 | 10 | 20|
> |-----------|-----------|-----------|-----------|-----------|
> | A-MWGraD-SVGD | 92.8 | 94.3 | 94.1 | 94.5|
> | A-MWGraD-Blob | 92.2 | 94.2 | 94.6 | 94.1|
>
> **Multi-Fashion**
>
> | K | 2 | 5 | 10 | 20|
> |-----------|-----------|-----------|-----------|-----------|
> | A-MWGraD-SVGD | 84.6 | 86.3 | 86.2 | 86.1|
> | A-MWGraD-Blob | 85.1 | 86.4 | 86.2 | 86.6|
>
> When $K=2$, the performance decreases. Increasing beyond $5$ provides limited improvement while significantly increasing training cost, so $K=5$ offers a good trade-off.
>
>
> >3. Can you clarify the baseline list in Section 4.2 and Table 1?...
>
> We thank the reviewer for pointing out this inconsistency. The intended baselines are:
>
> - MT-SGD, an SVGD-based multi-target sampling method that generates diverse particles in the joint high-density region shared by all targets.
>
> - MOO-SVGD, which incorporates multi-objective optimization into the SVGD framework to encourage particle diversity across targets.
>
> - MWGraD, which performs multi-objective optimization in probability space without acceleration.
>
> In our implementation, MWGraD with SVGD gradient approximation corresponds to MT-SGD, which is why the table reports this method as MWGraD-SVGD. MGDA is a classical multi-objective baseline in Euclidean space and can be applied to Bayesian multi-task learning by training separate models. However, prior experiments (Nguyen et al) show that MWGraD variants outperform MGDA in this setting, so MGDA is not included in our comparisons.
>
> We have revised Section 4.2 to clarify these relationships and avoid ambiguity.
>
>
> > 4. Can you soften or better substantiate the claim that A-MWGraD variants "consistently outperform" other methods? ...
>
> Our intention was to highlight that the accelerated variants often achieve faster convergence or better performance in several settings, rather than uniformly dominating all baselines.
>
> In the revision, we modify this claim as follows: "A-MWGraD variants often improve convergence compared to unaccelerated ones while achieving competitive or superior performance in several experiments."
>
> We'd like to thank the reviewer again for evaluating our paper. We hope that our responses/revision have addressed your concerns. Please let us know if you have any more questions.

---

> > ### Author Rebuttal · Reviewer_dJwV · 2026-04-01
> >
> > Thanks to the authors for their responses and for additional ablation experiments. I admit that theoretical analysis of particle system approximation is a large problem outside the scope of the current paper. I believe that the ablation study on $\sigma$ and $K$ provides enough empirical justification

---

> > > ### Author Response · Authors · 2026-04-04
> > >
> > > We'd like to thank the reviewer again for evaluating our paper.

---

### Official Review · Reviewer_uTT5 · 2026-03-13

**Soundness:** 3
**Presentation:** 2
**Significance:** 2
**Originality:** 2
**Overall Recommendation:** 4
**Confidence:** 4

**Summary:**

A previous work (Nguyen 2025) introduced multi-objective Wasserstein gradient flow (termed MWGraD by the authors) for optimizing multiple objectives in a direct extension of the framework of Euclidean multi-objective optimization (MOO) to the Wasserstein (distributional) setting (which the authors term MODO). The authors of the present work combine this result with the generalization of Nesterov’s momentum to the 2-Wasserstein space established in the accelerated Wasserstein gradient flow (Wang & Li 2022) to introduce A-MWGraD or an accelerated Wasserstein gradient flow for MODO. The authors show $\mathcal{O}(1/t^{2})$ convergence of this algorithm for geodesically convex objectives. They also introduce a kernel discretization and show the accelerated variant improves MWGraD in convergence speed and sampling efficiency.

**Compliance With Llm Reviewing Policy:**

Affirmed.

**Final Justification:**

The rebuttal mostly addressed my concerns. Though I have some reservations about the novelty of the work, its practical value/implications for ML, and a few technical details about the work (see review/rebuttal), I lean towards a weak accept as the work is essentially technically correct and the writing is clear.

**Key Questions For Authors:**

- The empirical evaluation demonstrates that A-MWGraD converges faster than MWGraD, yet the implementation essentially reduces to SVGD with momentum. Thus, a natural question to ask is how this technique compares to other accelerated sampling approaches or existing momentum-based SVGD variants, rather than just the unaccelerated baseline. Could the authors add a comparison to a few other accelerated variants?

- Given that the score is intractable for empirical measures under consideration, which yield the Blob and SVGD approximations as a result, to what degree do the Pareto theoretical guarantees transfer to the A-MWGraD algorithm as practically implemented?

**Limitations:**

yes

**Strengths And Weaknesses:**

Strengths:

- The theoretical results are sound. The authors of the present work take the limit of step-size $\eta \to 0$ to define what they term “MWGraD flow” from the work of (Nguyen 2025) and show that the merit function (e.g. similar to that of multi-objective optimization but for distributions) converges under a few generalizations of the Euclidean MOO to the Wasserstein setting. In particular, they assume $\beta$-strongly geodesically convex $F_{k}(\rho)$ functionals and use the Wasserstein distance as a definite Lyapunov function in Wasserstein-space in a standard argument to show convergence of the MWGrad flow at rate $\mathcal{O}(1/t)$, and then add a Nesterov acceleration term and apply a similar argument to show a convergence rate of $\mathcal{O}(1/t^{2})$.

Weaknesses:

- The generalization of Nesterov’s momentum to the 2-Wasserstein space, was established in the accelerated Wasserstein gradient flow of Wang & Li 2022. In the setting of what the authors call “MODO” (multi-objective distributional optimization, in parallel to “MOO” or multi-objective optimization), the authors seek the Pareto optimal with respect to multiple objectives over distributions. However, this generalization considered by the authors is relatively incremental given that multi-distribution optimization is well-explored, that MODO has been applied to WGF in a previous work in MWGraD (Nguyen 2025), and given that Nesterov acceleration has been generalized to accelerating WGF. In other words, extending Nesterov acceleration to the specific case of MWGraD is straightforward and incremental, and the bounds are direct.

- The implementation is a discrete particle discretization (i.e. supported on deltas), which implies one cannot compute the Wasserstein gradient for Bayesian sampling. The authors then approximate this with SVGD (Liu and Wang 2016) and Blob methods (Carrillo 2019), for which the continuous $W_2$ convergence arguments no longer strictly apply. Ultimately, the empirical results show that adding momentum to SVGD results in faster convergence, which is not exactly a surprising breakthrough.

- The empirical validation is performed on simplified and low-dimensional datasets, namely 2D toy Gaussian mixtures and MNIST/fashion MNIST variants. The practical utility for realistic high-dimensional distributional optimization problems is not especially clear.

---

> ### Author Rebuttal · Authors · 2026-03-27
>
> We thank the reviewer for the constructive comments. We will address the points raised below.
>
> > The generalization of Nesterov’s momentum to the 2-Wasserstein space, was established in the accelerated Wasserstein gradient flow of Wang & Li 2022. In the setting of what the authors call “MODO” (multi-objective distributional optimization, in parallel to “MOO” or multi-objective optimization), the authors seek the Pareto optimal with respect to multiple objectives over distributions. However, this generalization considered by the authors is relatively incremental given that multi-distribution optimization is well-explored, that MODO has been applied to WGF in a previous work in MWGraD (Nguyen 2025), and given that Nesterov acceleration has been generalized to accelerating WGF. In other words, extending Nesterov acceleration to the specific case of MWGraD is straightforward and incremental, and the bounds are direct.
>
> We believe that adapting the proof techniques from single-objective to multi-objective optimization, and from Euclidean space to probability space, is highly non-trivial. We have added a discussion of these challenges in the revision. We also refer the reviewer to our responses to Reviewer ULBH for additional discussion of the difficulties arising in the multi-objective setting.
>
>
> > The empirical evaluation demonstrates that A-MWGraD converges faster than MWGraD, yet the implementation essentially reduces to SVGD with momentum. Thus, a natural question to ask is how this technique compares to other accelerated sampling approaches or existing momentum-based SVGD variants, rather than just the unaccelerated baseline. Could the authors add a comparison to a few other accelerated variants?
>
> We would like to clarify the following important points. In this work, we focus on multi-objective optimization over probability space. To the best of our knowledge, **momentum-based SVGD methods have not been developed for multi-objective setting**. The only existing SVGD-based methods for multi-objective setting are proposed in [1] and [2], referred to as MOO-SVGD and MT-SGD, and their accelerated versions have not been studied. Furthermore, MWGraD [3], which is unaccelerated version of our method, reduces to MT-SGD (or SVGD in multi-objective setting) when we use SVGD to approximate the Wasserstein gradients of objective functions. In our experiments, we include MOO-SVGD and MWGraD-SVGD as baselines for comparison.
>
> [1] Liu et al., Profiling Pareto front with stein variational gradient descent, NeurIPS 2021.
>
> [2] Phan et al., Stochastic Multiple Target Sampling Gradient Descent, NeurIPS 2022.
>
> [3] Nguyen et al., Accelerated Multiple Wasserstein Gradient Flows for Multi-objective Distributional Optimization, UAI 2025.
>
> > Given that the score is intractable for empirical measures under consideration, which yield the Blob and SVGD approximations as a result, to what degree do the Pareto theoretical guarantees transfer to the A-MWGraD algorithm as practically implemented?
>
> In this work, we focus on the continuous-time convergence analysis of the accelerated method A-MWGraD. The discrete-time analysis for empirical measures remains open. However, we believe that establishing the continuous time analysis makes an important step toward a complete theoretical understanding of accelerated methods for multi-objective optimization over the probability space.
>
> We'd like to thank the reviewer again for evaluating our paper. We hope that our responses have addressed your concerns. Please let us know if you have any more questions.

---

> > ### Author Rebuttal · Reviewer_uTT5 · 2026-04-03
> >
> > I thank the authors for their response. I still believe that the points/concerns noted by reviewer ULBH and some of those covered in my original review above need to be addressed more thoroughly in the revised version of the paper. In particular, to add to ULBH's point on differentiability of the transport map I agree that the authors should state the appropriate regularity assumptions -- e.g., citing Caffarelli’s regularity theory and specifying conditions such as Holder $C^{k,\alpha}$ continuous density on $C^2$ domain to guarantee Schauder bootstrapped $C^{k+2,\alpha}$ Kantorovich potential -- to justify smoothness. Adding assumptions such as these would make the theoretical claims more complete and rigorous.
> >
> > However, trusting that the authors will implement the full scope of the required changes, I will raise my score to a "weak accept."

---

> > > ### Author Response · Authors · 2026-04-04
> > >
> > > We'd like to thank the reviewer again for evaluating our paper.

---

### Decision · Program_Chairs · 2026-04-30

**Decision:**

Accept (regular)

**Comment:**

This paper studies a vector-valued / multi-objective optimization problem over probability distributions.  (The notion of optimality is to be understood in terms of Patero optimality.)  A recent work towards solving this problem is the Multiple Wasserstein Gradient Descent (MWGraD) algorithm, which combines the idea of regular gradient descent over the Wasserstein space and some aggregation step that combines information across different objectives.  The specific contribution of this paper is an accelerated variant of the MWGraD algorithm.  The authors establish convergence rates for geodesically convex objectives.  Numerical experiments show improvements over the un-accelerated variant (MWGraD).

The general consensus among the reviewers is that the acceleration is performed in a principled way, and that the analysis is relevant and sound.  (The reviewers also commented that the paper is well-written.)  We are happy to accept the paper.

That said, the Reviewers do raise a number of useful suggestions including (i) regularity of optimal transport maps, (ii) a more in-depth discussion with larger datasets, and (iii) comparison with other accelerated variants.

It is hoped that the authors do take these suggestions into consideration when revising the paper.